# Towards General Loop Invariant Generation: A Benchmark of Programs with Memory Manipulation

**Chang Liu**[*], **Xiwei Wu**[*], **Yuan Feng, Qinxiang Cao**[‡]**, Junchi Yan**[‡]
Dept. of CSE & School of AI & Moe Key Lab of AI, Shanghai Jiao Tong University
{only-changer, yashen, fy0123, caoqinxiang, yanjunchi}@sjtu.edu.cn
Official Repository: https://github.com/Thinklab-SJTU/LIG-MM

## Abstract

Program verification is vital for ensuring software reliability, especially in the context of increasingly complex systems. Loop invariants, remaining true before and after each iteration of loops, are crucial for this verification process. Traditional provers and machine learning based methods for generating loop invariants often require expert intervention or extensive labeled data, and typically only handle numerical property verification. These methods struggle with programs involving complex data structures and memory manipulations, limiting their applicability and automation capabilities. In this paper, we introduce a new benchmark named LIG-MM, specifically for programs with complex data structures and memory manipulations. We collect 312 programs from various sources, including daily programs from college homework, the international competition (SV-COMP), benchmarks from previous papers (SLING), and programs from real-world software systems (Linux Kernel, GlibC, LiteOS, and Zephyr). Based on LIG-MM, our findings indicate that previous methods, including GPT-4, fail to automate verification for these programs. Consequently, we propose a novel LLM-SE framework that coordinates LLM with symbolic execution, fine-tuned using self-supervised learning, to generate loop invariants. Experimental results on LIG-MM demonstrate that our LLM-SE outperforms state-of-the-art methods, offering a new direction toward automated program verification in real-world scenarios.

## 1 Introduction

Program verification (1; 2; 3; 4) has become increasingly crucial due to the growing complexity of software systems. As software permeates various aspects of modern life, from critical infrastructure to daily applications, ensuring its correctness and reliability is paramount (5). Program verification aims to provide formal assurances that software behaves as intended, which is especially important in safety-critical systems such as autonomous vehicles (6), medical devices (7), and financial systems (8). Moreover, with the advent of artificial intelligence, the integration of verified components ensures the robustness and dependability of AI-driven solutions (9; 10). As software complexity and the integration of AI grows, so does the need for sophisticated verification tools to mitigate potential risks, making program verification an indispensable aspect of software engineering and the AI industry.

A crucial element in the realm of program verification is the concept of loop invariants (11; 12). **Loop invariants** are conditions that hold *true* before and after each iteration of a loop, serving as fundamental properties that ensure the correctness of loops within programs. Their importance cannot be overstated, as loops are ubiquitous and often complex in real-world software. By establishing loop invariants, developers can automate the verification process, significantly reducing the manual

---

*Equal contribution; ‡Corresponding authors. This work was supported by NSFC (92370201, 62222607).

38th Conference on Neural Information Processing Systems (NeurIPS 2024) Track on Datasets and Benchmarks.

effort required to prove the correctness of loops. This automation not only enhances the reliability of software but also increases efficiency by enabling rigorous analysis of program variables within loops. To better illustrate loop invariants, let's consider a simple case used in previous papers ([13]):

$$\frac{\{Pre\} \Rightarrow \{Inv\}; \quad \{Inv \wedge B\}\, S\, \{Inv\}; \quad \{Inv \wedge \neg B\} \Rightarrow \{Post\}}{\{Pre\}\ \textbf{while}\ B\ \textbf{do}\ S\ \{Post\}}$$

(a) Rule of loop invariant.

```
1 x := -50;
2 while (x < 0) {
3     x := x + y;
4     y := y + 1;
5 }
6 assert(y > 0)
```

(b) An example numerical program.

(c) A desirable loop invariant $I$ is a predicate over $x, y$ such that:

$$\forall x, y: \begin{cases} \text{true} \implies I[-50/x] \\ I \wedge x < 0 \implies I[(y+1)/y, (x+y)/x] \\ I \wedge x \geq 0 \implies y > 0 \end{cases}$$

(d) The desired loop invariant is $(x < 0 \vee y > 0)$.

Figure 1: A numerical program with a correctness assertion and a loop invariant to prove it.

To prove the correctness of this numerical program[1] in Fig. 1(b), researchers typically adopt symbolic execution systematically exploring codes to derive assertions (conditions). Starting from the beginning, we can record the assertions after conducting every program line. After line 1, the assertion becomes $x == -50$. However, the situation is complicated regarding the loop (line 2-5) because we do not know how the program behaves in loops. Loop invariant is used to fill this gap, and the rule of loop invariant is shown in Fig. 1(a). Via these rules, we find $(x < 0 \vee y > 0)$ satisfied the properties of loop invariant for this loop. Then, we can use the loop invariant to check the assertions after the loop: $(x < 0 \vee y > 0) \wedge \neg(x < 0) \Rightarrow (y > 0)$ (line 6). Finally, we reach the end of the program, and the assertion derived from the beginning is $y > 0$ as expected, which denotes our proof is successful.

Other than simple numerical programs, most real-world software applications involve the use of complex data structures and intricate memory manipulation. These include linked lists, trees, hash-tables, and various user-defined data structures. Verifying programs that operate on these complex data structures is significantly more challenging than dealing with numerical programs. The complexity arises from the need to reason about the shape and content of data structures, manage aliasing and pointer dereferencing, and ensure the integrity of dynamic memory. Moreover, such programs often exhibit intricate behaviors and dependencies that make the inference of loop invariants particularly demanding, which can be shown by the following example in Fig.2(Left):

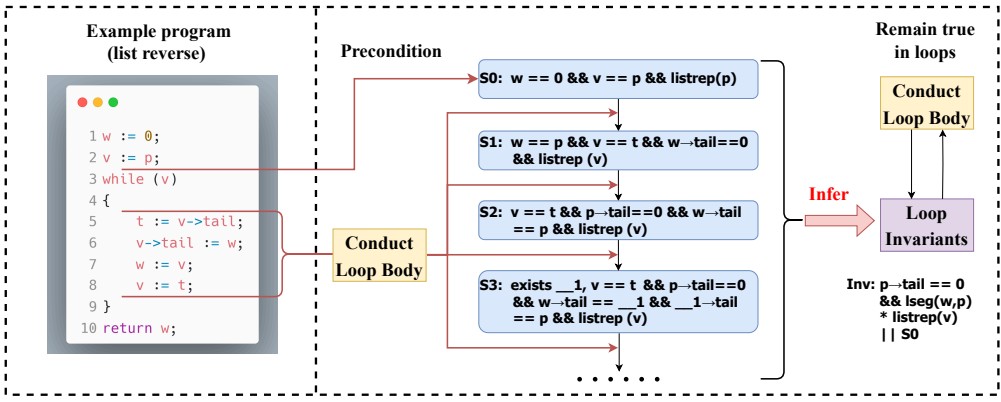

Figure 2: (Left) An example of programs with the data structure like the single linked list; (Right) The pipeline of our proposed LLM-SE: we start with the precondition (S0) and conduct the loop body with symbolic execution multiple times to get more separation logic assertions (S1, S2, S3, ...). Based on these separation logic assertions, we further use LLM to infer the loop invariant.

---

[1]Numerical programs refer to the programs that do not contain any data structures or memory manipulation.

To prove the correctness of the program *list reverse* manually in Fig. 2(Left), researchers also use symbolic execution to determine the assertions that the program states must satisfy. First, we need to define some predicates to describe the memory states. For example, we use **listrep(x)** to denote a singly linked list starting from pointer x and **lseg(x,y)** to denote a singly linked list segment from the pointer x to the pointer y (excluding y). To simplify assertions, researchers introduced separation logic(14) instead of relying solely on traditional first-order logic. In separation logic, $P * Q$ indicates that $P$ and $Q$ describe properties of two disjoint memory segments. This approach eliminates the need for numerous auxiliary propositions to specify which addresses are distinct. With this groundwork in place, we can easily provide a description of the program state before entering the loop (line 2): `w == 0 && v == p && listrep(p)`. After reading the codes, we find this loop reverses the singly linked list pointed by v and store it in w. Therefore, experienced researchers can infer the loop invariant `w == 0 && v == p && listrep(p) || p → tail == 0 && lseg(w,p) * listrep(v)`. After the loop ends (line 9), v is a null pointer and w points to the reversed linked list. Therefore, the final program state is represented by `listrep(w)`, which denotes that we have completed the verification of this program's memory safety. In this example, because we have a clear understanding of how the program manipulates memory, we can easily write the loop invariant. However, as the number of function calls increases and the memory involved becomes more complex, it is not always straightforward to comprehend the algorithm and write the loop invariant manually.

Traditional program analysis works have made some attempts at automatic loop invariant generation, including LOOPINVGEN (15; 16), llinva (17), and SLING (18; 19). However, they rely heavily on fixed templates for feature extraction, which makes them less adaptable. Techniques like synthesis, abduction, and dynamic analysis require expert intervention and the design of specific inputs. Machine learning approaches have also been explored for loop invariant generation (20; 21; 22; 23; 24; 25; 13; 26; 27). Nevertheless, in this task, the labeled data require valid loop invariants, which are hard to automatically generate without expert intervention. In one foundational work CODE2INV (13), they use reinforcement learning to avoid label sparsity. Nevertheless, such approaches require much trial and error, which has to be performed on the target programs directly and cannot be pre-trained offline in advance. Another paradigm (25; 20) is to borrow the loop invariants found by traditional methods as labels. While the trained models can hardly surpass traditional methods, they can only offer limited speed advantages. Recently, several works (25; 27; 20) have begun to utilize large language models in this task. However, to the best of our knowledge, **existing machine learning based methods are all restricted to program's numerical property verification**, neglecting shape analysis and memory safety verification for real-world programs with diverse data structures and memory manipulation.

Unlike existing works, we believe that automatic verification for programs with sophisticated data structures and complex memory manipulation is necessary and significant. In this paper, we propose a **L**oop **I**nvariant **G**eneration benchmark of programs with **M**emory **M**anipulation (**LIG-MM**), which is more challenging than the numerical programs in previous works. LIG-MM contains 312 programs collected from various sources, including common programs from course homework, programs from the international competition on software verification (SV-COMP(28)), programs from previous papers (SLING(18; 19)), programs from real-world software and systems(Linux Kernel(29), GlibC(30), LitesOS(31), and Zephyr(32)). Our benchmark requires the ability of methods to perform shape analysis and memory safety verification via generating valid loop invariants.

We have evaluated multiple baselines on our proposed LIG-MM benchmark, including state-of-the-art traditional prover, machine learning based methods, and the popular large language model GPT-4. Though these baselines may perform well on numerical programs, they can hardly work on our LIG-MM, as their pass rates are less than 15% in one attempt. It has demonstrated the challenges of our LIG-MM compared to existing numerical program benchmarks, and the difficulty of generating loop invariants reaches a new level when it involves data structures and memory manipulations.

To address the challenge in our LIG-MM benchmark, we further propose a new framework combining **L**arge **L**anguage **M**odels and **S**ymbolic **E**xecution named **LLM-SE**. As shown in Fig. 2(Right), by integrating symbolic execution, we can systematically explore program codes and generate separation logic assertions sufficient for LLM to infer loop invariants. This hybrid approach not only automates the verification process but also significantly reduces the need for expert intervention and tailored inputs. Furthermore, our method is scalable to multi-layered loops and multi-lever pointer operations, allowing for efficient and effective invariant generation across various program types.

To sum up, our main contributions are:

1. We construct a challenging benchmark named LIG-MM of separation logic-based loop invariant generation to verify the shape safety of memory-manipulated programs.

2. We evaluate multiple baselines including GPT-4 on our dataset to analyze the capacity of existing approaches to generating loop invariants in our LIG-MM benchmark.

3. Aiming at the LIG-MM benchmark, we propose a new approach LLM-SE that combines large language models with symbolic execution, and demonstrate its performance. We believe it will be a new direction for loop invariant generation and the entire program verification field.

## 2  Background

### 2.1  Separation Logic Assertion

Separation logic (14) is a formal system introduced by John C. Reynolds in the early 2000s, which is for reasoning about the correctness of computer programs. Separation logic uses a new connective separating conjunction to ensure that different names duplicate no identical addresses. The separating conjunction $P * Q$ represents the existence of two disjoint portions of the state, one that satisfies $P$ and one that satisfies $Q$. Specifically,

$$m \models P * Q \iff \exists m_1, m_2 . m = m_1 \uplus m_2, m_1 \models P \&\& m_2 \models Q.$$

Here $\uplus$ means the disjoint union[2] and $m \models P$ means $m$ satisfies $P$. A distinction between $*$ and $\&\&$ is that $P * P \neq P$ where $P \&\& P = P$. In particular, if **store**$(p, v)$ means that the value $v$ is stored at address $p$, **store**$(p, v) *$ **store**$(q, u)$ implies $p \neq q$, and thus **store**$(p, v) *$ **store**$(p, v)$ is always false: there is no way to divide a heap in such a way that a cell $p$ goes to both partitions.

Therefore, we can define separation logic assertions used to describe program states, which take the form $\exists \vec{x} (P_1 \&\& P_2 \&\& \cdots \&\& P_n \&\& Q_1 * Q_2 * \cdots * Q_m)$, where the pure part $P$ describes properties between terms (e.g. e1, e2), and the spatial part $Q$ is a separating conjunction of spatial predicates. For example, e1==e2 and e1>e2 can appear as pure propositional conjuncts; the empty heap predicate emp, the points-to predicate store(e1,e2) can appear as spatial conjuncts. Based on the definitions of emp and store, we can give specific definitions of listrep and lseg.

```
listrep(x) := x == 0 && emp || ∃ z, store(&(x → tail), z) * listrep(z)

lseg(x,y)  := x == y && emp || ∃ z, store(&(x → tail), z) * lseg(z, y)
```

### 2.2  Symbolic Execution and Correctness Check of Loop Invariant

Symbolic execution is a program analysis technique that explores the execution paths of a program symbolically, without concrete inputs(33). Put simply, symbolic execution is a function that takes assertions and program statements as input and evaluates an assertion. It calculates the strongest postcondition after each statement. The main usage of symbolic execution is to generate separation logic assertions. Take the example in Fig. 2(Left & Right) for an example:

We start the symbolic execution process before entering the loop body (line 2) at the precondition S0: w == 0 && v == p && listrep(p). When entering the loop, as the loop condition (line 3) becomes true, the assertion becomes w == 0 && v != 0 && v == p && listrep(p). Based on the definition of the predicate **listrep**, we will expand this assertion into an equivalent form ∃ v0, w == 0 && v == p && p → tail == v0 && listrep(v0), which guarantees the legitimacy of subsequent reads and writes to the v-address. After four assignment statements (line 5-8), the assertion becomes S1: w == p && v == t && w → tail== 0 && listrep(v). We have calculated the strongest postcondition S1 for S0 after one cycle of the loop body. Similarly, we can get the strongest postcondition S2 S3 or more (the blue boxes of Fig. 2(Right)) for executing the loop body several times. Via symbolic execution, we can acquire sufficient separation logic assertions, which may be useful in inferring loop invariants.

Moreover, we can complete the correctness check of loop invariants via symbolic execution. As shown in Fig. 1(a), a legitimate loop invariant I needs to satisfy three properties (1) I is true before

---

[2] $A \uplus B = A \bigcup B$ if $A \bigcap B = \emptyset$, e.g., $\{1, 2, 3, 5\} = \{1, 3\} \uplus \{2, 5\}$. But $\{1, 3, 5\} \uplus \{1, 2, 3\}$ is undefined.

the first iteration of the loop (2) if `I` is true before an iteration, it remains true before the next iteration (3) `I` will not generate error after the end of the loop. For the first property, we need to check that the pre-condition of the loop derives `I`. For the third property, we simply go on to symbolic execution with `I` $\land \neg$**condition**. As for the second property, we need to compute the strongest postcondition of `I` after executing the loop body once and check if it derives `I`. After all these three properties are checked by the symbolic execution, we can conclude that the loop invariant is correct.

## 2.3 Related Work

The related works cover different aspects of works that are related to our work. Due to the page limit, we have to place it in the Appendix (A.1).

## 3 LIG-MM: Loop Invariant Generation Benchmark of Programs with Memory Manipulation

As we mentioned before, the benchmark programs in existing papers mostly contain numerical programs. To fill the lack of benchmarks for general loop invariant generation, we propose LIG-MM, a loop invariant generation benchmark of memory manipulation programs. Table 1 below shows the basics of the code in LIG-MM. Our programs come from four main sources: course codes, competition codes, previous relevant work, and the actual system codes. The programs are modified into a unified format for better usage. Multiple examples are shown in the Appendix (A.3), and the licenses of existing benchmarks can also be found in the Appendix (A.5).

- *Course codes.* The course code is mainly derived from homework programs on the data structure course and programming language course. The detailed course number and college name are covered due to the privacy requirements. These programs contain the most diverse data structures and multi-level pointer operations among the sources of our benchmark.

- *Competition codes.* SV-COMP(34; 28) is a competition on software verification, which provides a benchmark for verification tasks for comparing verification tools. Originating from competition, this dataset encompasses various verification tasks, providing a comprehensive set of real-world and synthetic examples for testing the effectiveness and efficiency of verification techniques. In our LIG-MM, we select programs from the 2022 competition benchmark.

- *Previous relevant work.* SLING (18; 19) uses traditional dynamic analysis techniques to generate invariants. Other than loop invariants, SLING can also generate preconditions and post-conditions for function. Therefore, not all their benchmarks include the inference of loop invariant or even contain a loop (they use function calls to replace loops). After selection, we choose some of the programs in its benchmark and turning them into a uniform code style.

- *Real-world system codes.* To make the data in LIG-MM closer to real-world software environments, we decide to select more programs from several well-known software and systems. Among them, GlibC (30) is the GNU implementation of the C standard library, providing essential functionalities for numerous applications. Additionally, we have incorporated programs from the Linux Kernel (29), a widely used and highly-regarded operating system kernel that serves as the foundation for countless devices and systems worldwide. To further enhance the diversity of our dataset, we have included LiteOS (31), a lightweight operating system designed for IoT devices, and Zephyr (32), another versatile operating system known for its applicability in resource-constrained environments.

By integrating these varied sources, LIG-MM captures a broad spectrum of programming practices and challenges and ensures that our benchmark is robust and representative of the complexities encountered in multiple scenarios, such as real-world software development and verification. Unlike the numerical program benchmark in previous works (15; 16; 13; 25; 27; 20), **our benchmark does not contain pure numerical procedures, all of our programs are related to at least one certain data structure.** The data structures we have selected include singly linked lists(sll), doubly linked lists(dll), trees, and hash-tables. In addition, our benchmark includes the usage of multi-level pointers and various pointer arithmetic.

Table 1: Statistics of our proposed LIG-MM benchmark.

| | Concrete Resources | Number of Programs | Data Structure Types |
|---|---|---|---|
| Course codes | Course homework programs | 187 | sll, dll, tree, hash-table |
| Competition codes | SV-COMP (34; 28) | 27 | sll, dll, tree, hash-table |
| Previous benchmark | SLING (18; 19) | 15 | sll, dll, tree |
| Real-world programs | Linux Kernel (29) | 23 | sll, dll, hash-table |
| Real-world programs | GlibC (30) | 13 | dll, hash-table |
| Real-world programs | LiteOS (31) | 12 | dll |
| Real-world programs | Zephyr (32) | 35 | sll, dll, hash-table |
| Overall | - | 312 | sll, dll, tree, hash-table |

# 4 LLM-SE: Loop Invariant Generator via Coordinating Large Language Models and Symbolic Execution

As we mentioned before, the core idea of our approach is to infer proper loop invariants based on the separation logic assertions provided by symbolic execution. In this section, we first explain why we believe the LLM is supposed to work in the task of loop invariant generation. Then, we briefly introduce the preliminaries of the symbolic execution used in our work. Finally, we propose a framework to auto-generate loop invariants, combining the power of LLM and symbolic execution introduced in Sec 2.2.

## 4.1 Insights of Inferring Loop Invariants

As the basis of inferring, we plan to use symbolic execution to generate sufficient separation logic assertions. Since they are derived by the same loop body after the precondition, several patterns or similarities should exist inside them, which can be captured for inferring. To take a deeper look, we use the list reverse program in Fig. 2 as an example:

```
S0: w == 0 && v == p && listrep(p)
S1: w == p && v == t && w → tail == 0 && listrep(v)
S2: w → tail == p && v == t && p → tail == 0 && listrep(v)
S3: exists__1, w → tail == __1 && __1 → tail == p &&
    v == t && p → tail == 0 && listrep(v)
S4: exists__1__2, w → tail == __1 && __1 → tail == __2 && __2 → tail == p &&
    v == t && p → tail == 0 && listrep(v)
S5: exists__1__2__3, w → tail == __1 && __1 → tail == __2 && __2 → tail == __3
    && __3 → tail == p && v == t && p → tail == 0 && listrep(v)
```

In this example, we write down their separation logic assertions after multiple symbolic executions. We can find several terms in [S0,S1,S2,S3,S4,S5,...] that are related to lseg(w,p), and we mark these terms with red. Since the position and the order of these terms may vary, one may need to design a specific algorithm to detect such a syntactic pattern in traditional program logic research. As we all know, LLM has strong abilities in extracting features and detecting patterns, and thus, it has the potential to infer loop invariants or some key parts of them. Supposing LLM successfully infers lseg(w,p), then we can modify the original separation logic assertions by replacing those red terms:

```
S0:  w == 0 && v == p && listrep(p)
S1': v == t && p → tail == 0 && lseg(w, p) * listrep(v)
S2': v == t && p → tail == 0 && lseg(w, p) * listrep(v)
S3': v == t && p → tail == 0 && lseg(w, p) * listrep(v)
S4': v == t && p → tail == 0 && lseg(w, p) * listrep(v)
S5': v == t && p → tail == 0 && lseg(w, p) * listrep(v)
```

Moreover, we can find that other terms in the updated separation logic assertions also have patterns. For example, p→tail==0 appears in S5', S4', S3', S2', and S1'. As a result, the LLM can continue to infer p→tail==0, update the separation logic assertions again and further infer listrep(v). Now, every term in the separations logic assertions has been covered by the terms inferred by LLM, except for S0. Finally, we get the following loop invariant candidate.

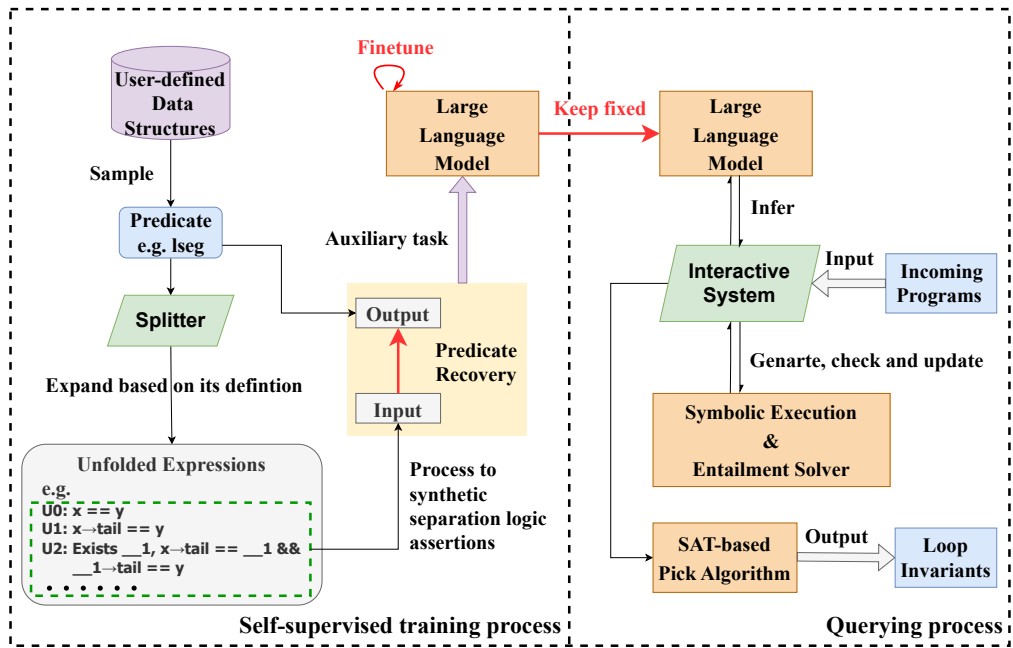

Figure 3: Our proposed framework LLM-SE. (Left) Offline training: we construct an auxiliary task by splitting and recovering the predicates of data structures, following the self-supervised learning paradigm to finetune LLM; (Right) Online querying: we design an interactive system to handle the programs needed to be verified. The well-trained LLM is directly applied to unseen programs. Multiple verification tools are utilized to cooperate with the LLM to generate valid loop invariants.

```
inv: p→tail==0 && lseg(w,p) * listrep(v) || w == 0 && v == p && listrep(p)
```

By constantly inferring the terms of loop invariants via LLM, we can end up with valid loop invariants. Please note that this is only a very simple example and many programs are way more complicated than this one, of which the loop invariant is not easily found. We will place more examples in the supplementary file, please refer to them if interested.

## 4.2 Framework Overview

In this work, we propose to combine the power of pretrained LLM and traditional verification tools, aiming at efficiently inferring the loop invariants. The overall framework of LLM-SE is shown in Figure 3, including the offline training process and the online querying process, where offline training involves fine-tuning LLM, and online querying refers to the LLM's real-time usage for generating loop invariants of incoming programs unseen before.

In the offline training process, the major challenge is the scarcity of labeled data. In loop invariant generation, the data refers to programs and the label refers to valid loop invariants, which require expert domain knowledge and are hard to produce massively. To address this issue, we decided to borrow the power of self-supervised learning, to create enough labeled data for training by designing an auxiliary task. As Figure 3(left) shows, we generate extensive labeled data by splitting predicates of data structures based on their definitions, and treat the reassembly of these split predicates as the auxiliary task. Furthermore, we employ *data augmentation* and *mix up* techniques to further process the synthetic data and leverage the difficulty of the auxiliary task. This process is totally automatic, and can be readily applied to new data structures and predicates, as long as they are well-defined.

The online querying process, illustrated in Figure 3(right), involves an interactive system between the LLM and multiple traditional verification tools, which allows the latter to validate and complete the inferences of the former, finally refining the outputs of LLM to valid loop invariants. In essence, LLM first infers a term based on current separation logic assertions generated by symbolic execution. Then, we check and update the separation logic assertions by replacing the related terms via the

entailment solver. This cycle persists until the terms of the original separation logic assertions are all covered, resulting in a collection of updated assertions, which could be composed of the loop invariants. Finally, we utilize an SAT-based pick algorithm to choose a concise and equivalent set of them to complete the loop invariant, ensuring validity and correctness. This collaborative approach merges the computational power of LLM with the rigorous validation and refinement capabilities of verification tools, and effectively yields a number of dependable loop invariants.

Due to the page limit, we have to move the detailed introduction of our proposed LLM-SE into the Appendix (A.2). Please refer to it for more details.

## 5 Experiments

In this section, we evaluate our proposed LLM-SE and other baselines on our LIG-MM. The content of our benchmark and the repository of our code will be illustrated in the supplementary materials, which allow researchers to reproduce the results shown in our experiments easily.

### 5.1 Set up

Our LLM-SE is implemented in Python and based on the HuggingFace [3] framework. We choose one pretrained LLM called CodeGen (35; 36) as backbone to propel our approach. It is tailored explicitly for code-related tasks with various applications (37; 38; 39), endowed with a comprehensive understanding of code structures, program semantics, and syntax. After several attempts, we found that the 350M version of CodeGen is very cost-effective and has considerable performance. With this model size, our LLM-SE framework can be easily adopted on common servers and PCs.

We conducted experiments on a Linux workstation with NVIDIA 3090 GPUs and an AMD Ryzen Threadripper 3970X 32-Core CPU with 128G RAM. In the offline process, we generated 200,000 synthetic data for the auxiliary task we mentioned before, and the model size after fine-tuning is merely 1.4 GB. In the online querying process, we directly feed the benchmark programs to the well-trained LLM-SE without further modification. Please note that the benchmark programs are totally unseen during offline training, and there is no need to worry about overfitting issues.

For the baseline, we compare our proposed method with the following methods: 1) **SLING** (18), the state-of-the-art method for separation logic loop invariant detection. We follow their official repository [4] to implement it, but it is worth noting that the docker provided is missing, and we create a similar workstation on our own; 2) **AutoSpec** (27), the latest LLM based method for loop invariant generation on numerical property verification, which is selected as a representative of existing numerical program based works; 3) **GPT-4** (40), the most commonly used LLM in research papers, and the prompt text we used in our experiments is shown in the Appendix (A.4).

For the use of our LIG-MM benchmark, the programs in LIG-MM are all regarded as the test set, and we do not provide the training set. The definitions of predicates are collected and passed to the methods in advance, where our LLM-SE uses them for self-supervised learning, GPT-4 uses them for prompt learning, SLING and AutoSpec just regard them as part of the domain knowledge.

Table 2: Experimental results on our LIG-MM benchmark, 312 programs in total. Pass rates are reported as Pass Rate @ N, where N is the number of attempts to generate loop invariants.

| Method | SLING (18) | AutoSpec (27) | GPT-4 (40) | LLM-SE (ours) |
|---|---|---|---|---|
| Pass Rate @ 1 | 12.82% | 0.00% | 12.50% | **38.78%** |
| Pass Rate @ 8 | 21.79% | 0.00% | 17.68% | **42.95%** |

### 5.2 Results and Discussion

Table 2 and Figure 4 show the experimental results on the LIG-MM benchmark. We can see that our proposed LLM-SE surpasses all baselines by a higher pass rate. The pass rate denotes the

---

[3] https://huggingface.co/
[4] https://github.com/guolong-zheng/sling

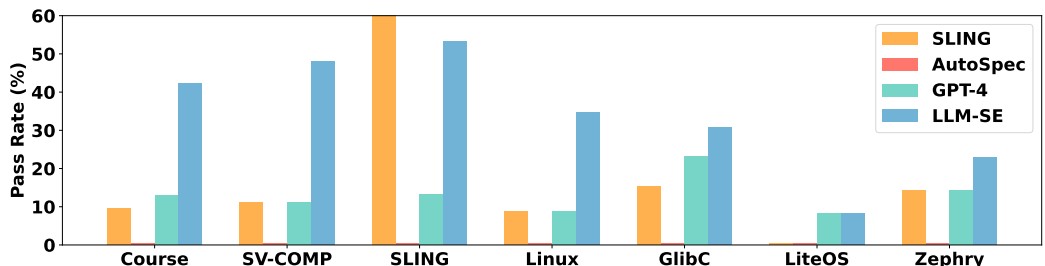

Figure 4: The one attempt pass rate (Pass Rate @ 1) on every source of programs in our LIG-MM, where the x-axis denotes the different sources and four bars represent the pass rates of four methods.

correctness of the output loop invariant. In our benchmark, we provide an entailment solver to check the correctness of the output loop invariant automatically (refer to Sec. 2.2). But for baselines other than our LLM-SE, the output loop invariant of their results may not always be in valid format, especially for GPT-4. Therefore, we have to manually check the correctness again for the ones in invalid format, which makes the pass rate of baselines higher to some extent.

We consider the poor performance of baseline AutoSpec to be due to the difference between numerical programs and memory-involved programs. AutoSpec is an outstanding framework for generating loop invariants for numerical property verification, but it turns out the special design and the prompt learning of AutoSpec on numerical programs do not work on our LIG-MM benchmark.

The performance of the traditional method of SLING is also not satisfactory. It may be because the programs used in the paper of SLING extensively use function calls instead of real loops. As listed in their official GitHub repository [5], most of their programs do not contain a loop. As a result, SLING mainly focuses on inferring pre/postconditions in these programs. In contrast, our benchmark programs all contain at least one loop, and several programs contain extra function calls to more loops. Moreover, SLING can hardly infer the correct loop invariants in one attempt, often requiring tens or hundreds of attempts in one program. This characteristic makes it perform poorly, especially on Pass Rate @ 1, as it failed on some of its own benchmarks.

Though our proposed LLM-SE outperforms all baselines, its pass rate may still not be sufficient to meet our expectations. We still need to improve the pass rate to make it applicable to the real world. We hope our work can pave the way for new advancements in loop invariant generation and inspire the broader community. We encourage more researchers to build upon our findings and further enhance the capabilities and reliability of automated program verification techniques.

Table 3: Ablation study results of different methods. We sequentially exclude key components from LLM-SE (ours): data augmentation (data aug.), self-supervised learning (SSL), and symbolic execution interaction (SE). Pass rates are reported as Pass Rate @ N, the same as before.

| Method | Pass rate @ 1 | Pass rate @ 8 |
|---|---|---|
| SLING | 12.82% | 21.79% |
| Auto-Spec | 0.00% | 0.00% |
| GPT-4 | 12.50% | 17.68% |
| LLM-SE (ours) | 38.78% | 42.95% |
| w/o data aug. | 27.88% | 32.05% |
| w/o SSL | 8.33% | 13.46% |
| w/o SE (CodeGen) | 0.00% | 0.00% |

[5]https://github.com/guolong-zheng/sling/tree/master/PLDI19_AE/pldi-sling/benchmarks

## 5.3 Ablation Studies

We conducted ablation studies by excluding different components of our LLM-SE. The key components sequentially removed include the data augmentation module (data aug.), the self-supervised learning module (SSL), and the interaction with the symbolic execution module (SE). Removing these components effectively degrades the model to simpler versions, with the final configuration being reduced to the base large language model, CodeGen.

Table 3 summarizes the results of the ablation study. We have evaluated the model's performance using the pass rate at 1 attempt and 8 attempts. The results show that the self-supervised learning (SSL) component is essential for our LLM-SE's performance, as the model significantly underperforms without it. In contrast, the data augmentation module has a more supplementary role, providing performance improvements but not as critical as SSL. As expected, the performance drops drastically when symbolic execution (SE) is removed, reducing the model to the base CodeGen, which fails to perform effectively in our tasks.

Furthermore, we conduct analysis the performance of these methods on different data structures. The pass rate @8 for each method is summarized in Table 4.

Table 4: Pass rate @ 8 on different data structures for various methods.

| Data Structure | SLING | AutoSpec | GPT-4 | LLM-SE (ours) |
|---|---|---|---|---|
| Single linked list | 33.33% | 0.00% | 34.38% | 61.46% |
| Double linked list | 24.81% | 0.00% | 13.53% | 32.33% |
| Listbox | 0.00% | 0.00% | 6.67% | 41.67% |
| Others | 13.04% | 0.00% | 8.70% | 30.43% |
| **Total** | 21.79% | 0.00% | 17.68% | 42.95% |

As shown in Table 4, our proposed LLM-SE demonstrates considerable performance across all types of data structures. In comparison, GPT-4 performs relatively well on the single linked list, while SLING shows better results on the double linked list. However, both methods struggle significantly with more complex or less common structures like the Listbox, where LLM-SE excels with a pass rate of 41.67%. This indicates that while baseline methods such as SLING and GPT-4 handle simpler structures to some extent, they fall short when it comes to intricate memory manipulations.

LLM-SE achieves a total pass rate of 42.95%, highlighting its robustness and flexibility in handling diverse data structures. In contrast, the inconsistent results from SLING and GPT-4, particularly on data structures beyond single and double linked lists, reveal the limitations of these methods. AutoSpec, with near-zero performance across all categories, underscores the need for significant improvements in handling diverse data structures within program verification tasks.

In summary, existing methods show varying levels of success with specific data structures, but their overall performance is inconsistent and inadequate for comprehensive program verification. Consequently, our proposed LLM-SE fills this gap by providing a more reliable and effective solution for managing a wide range of data structures.

## 6 Conclusion

In summary, we introduce a new loop invariant generation benchmark, LIG-MM, particularly for programs with complex data structures and memory manipulations. Due to the poor performance of existing methods on LIG-MM, we further propose a new framework, LLM-SE, which leverages LLM fine-tuned using a self-supervised learning approach and symbolic execution. Our experiments show that LLM-SE generates valid loop invariants more accurately and efficiently than the state-of-the-art baselines. Our LLM-SE accommodates various data structures, supports multi-loop scenarios, and simplifies the integration of new user-defined data structures. This makes our method a powerful alternative to traditional and reinforcement learning-based approaches. Our work highlights the potential of LLM in program verification and opens new avenues for research in this area.

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

# A  Appendix

The appendix is organized as follows: 1) Related work of our paper; 2) Details of our proposed framework LLM-SE; 3) More cases of programs in our proposed benchmark LIG-MM; 4) License of the programs we used; 5) The prompt text of GPT used in the baselines; 6) Discuss of limitation, impact, and outlook of our work.

## A.1  Related Work

Traditional approaches for program invariant generation rely on program synthesis and analyze. LoopInvGen is a data-driven tool that generates provably sufficient loop invariants for program verification (41). It transforms the loop invariant inference problem into a sequence of precondition inference problems, and solves them using a precondition inference engine (PIE (42)), which employs a program synthesis technique to learn features in a focused manner. Llinva (17) implemented an algorithm that automatically generates loop invariants using Why3 (43) and GPID (44). It generates verification conditions using Why3, and strengthens the expressions in the verification conditions using GPID and abduction techniques. Then, it passes them to an SMT solver to check their validity, and repeats this process until it obtains the loop invariants. By analysis the execution process for some test inputs, SLING (18) can automatically generate precondition, post-condition and loop invariants without other messages. But SLING only infers shape properties using inductive predicates and pure equality. These properties have strict patterns and they do not consider general disjunctive invariants or numerical relations. Crucially, it is very dependent on the sample inputs. It need smart test-input generation techniques to keep the correctness and efficiency. (45) adapts a different technique, which generates loop code from the loop invariants to learn specified algebraic relations among their terms, instead of others generating invariants from the loop body.

Thanks to the rapid development of machine learning, several learning-based approaches have been has been increasingly studied. ICE-DT  (21) utilizes decision trees over manually designed features, and intuitively uses one learner and one teacher for predicting invariants. CODE2INV (13) uses reinforcement learning (RL) with graph neural networks to train the agent for generating candidate loop invariants following the syntax and semantic. With the help of RL, they do not need labeled data with expert knowledge, while utilizes Z3 (46) to provide reward for training. Though it can outperform LOOPINVGEN and ICE-DT, the main drawback of using RL is that they need to directly train their agents on the evaluation sets with numerous trials and errors, which could lead to inefficiency. Following CODE2INV, several works have been proposed to improve it. (47) combines RL with an heuristic method called nondeterministic strategies, where RL is trained to guide the searching direction of the heuristic method. (48) expands CODE2INV on non-linear loop invariants via well-trained gated continuous logic networks, which improve the performance of CODE2INV significantly. CLN2INV (26) replace the graph neural networks used in CODE2INV to continuous logic networks and introduce a SMT-based tool to make it possible to generate loop invariants for real-world programs. (22) proposes to automatically construct inductive loop invariants for loop structures consisting of multiple paths inside, which generates loop invariants between forward and backward reachability of the loop. However, existing machine learning based methods for loop invariant generation can only handle numerical programs with scalar variables, which limits their usage since programs in the real world often include data structures and memory operations.

Recently, large language models (LLM) have emerged as a transformative force in the field of machine learning, particularly when applied to code-related tasks (35; 36; 49). The advent of models such as BERT (Bidirectional Encoder Representations from Transformers (50)) and GPT (Generative Pretrained Transformers (51)) has signified a turning point in the way we approach natural language understanding and generation. These models are pretrained on vast corpora of text, such as Wikipedia, books, and news articles. By learning from these diverse sources, LLM acquire a deep understanding of natural language, including its structure, context, and semantics. They use a special architecture called Transformer, which consists of multiple layers of self-attention and feed-forward layers. The Transformer architecture enables LLM to capture long-range dependencies and complex patterns during pre-training. After pretraining on large corpora of text, LLM can be fine-tuned on specific tasks and domains, such as text generation, classification, and summarization. By using LLM, we can leverage their powerful language understanding and generation abilities. This recent surge in LLM development has inspired a fresh wave of research in code-related tasks, ranging from code summarization, code completion, and code-to-code translation.

In our work to explore loop invariant generation, we draw upon the impressive strides made in LLM's ability to process and generate logic assertions, as these models promise to transcend the challenges posed by complex program semantics and syntax. The synergy between LLM and code-related tasks presents an exciting avenue for our proposed approach in this paper. Several recent work (20; 25; 27) also try to utilize LLM to solve the invariant generation task. One possible paradigm is to use traditional solvers such as Daikon (52) to generate valid invariants for the programs in the dataset, and regard the generated invariants as labels for training. In other words, they regard traditional solvers as the oracle and use LLM to approximate it. This approach has a major drawback that the LLM trained by traditional solver can hardly outperform the solver under the same conditions. In contrast, our self-supervised learning approaches LLM-SE do not have this limitation.

To the best of our knowledge, existing machine learning-based methods are restricted to verifying programs' numerical properties, neglecting shape analysis and memory safety verification for real-world programs with diverse data structures and memory manipulation. Therefore, we consider it necessary to propose a new benchmark to cover the programs with memory manipulation, as the LIG-MM benchmark proposed in our work.

## A.2 Details of our proposed framework LLM-SE

As mentioned in Sec 4, our proposed LLM-SE combines large language models and symbolic execution. Recall the overview in Fig. 3, LLM-SE includes two processes, the offline self-supervised training process and the online querying process. In the following section, we shall introduce these two processes. We suggest the reader understand Sec 4 and Fig. 3 first before reading the following documents.

### A.2.1 Offline Training with Self-supervised Learning

Our methodology relies on the usage of LLM, which is essential for solving the complex task of loop invariant generation. In practice, LLM after pretraining on large corpora of text can be fine-tuned on specific tasks and domains. We can leverage its powerful language understanding and generation abilities in loop invariant generation. After extensive investigation and experiments, we choose one pretrained LLM named CodeGen (35; 36) to propel our approach. It is tailored explicitly for code-related tasks with various applications (37; 38; 39), endowed with a comprehensive understanding of code structures, program semantics, and syntax. The unique prowess of CodeGen lies in its ability to decode and generate code snippets by comprehending the intricacies of different programming languages. Its robust capabilities allow us to delve into the task of loop invariant generation, presenting a promising avenue for addressing the complexities inherent in this task.

After selecting the pretrained LLM, we need to fine-tune it on our task. However, a major challenge is that we lack enough labeled data, i.e., programs with valid loop invariants. Without sufficient data for fine-tuning, the LLM cannot fully demonstrate its potential. To overcome this challenge, we adopt a self-supervised learning approach to generate abundant labeled data for fine-tuning our LLM. By formulating auxiliary tasks within the self-supervised framework, we generate rich synthetic data by employing a split-and-reassembly technique based on data structure definitions. Such a strategy allows us to produce ample training data, which is essential for fine-tuning LLM in the absence of labeled data, and thus addresses the challenge of data scarcity.

The detailed process of our self-supervised learning paradigm is shown in Figure 5. In this figure, we examine the "splitter" module of Figure 3, as the part enclosed by the green dash line. Given the data structures defined by the users, every time we sample one predicate from the data structures, e.g. `lseg`. Then, we check its definitions given by the users shown in the purple box:

```
lseg(x,y) = x == y && emp || ∃ z, x→tail == z * lseg(z,y)
```

We can see that there are two branches in its definition, one for the empty case and the other one for the non-empty case. We find that multi-branch definition is very common in data structures. Therefore, we decide to split the predicate and let the LLM try to reassemble it. This *predicate recovery* task is the auxiliary task designed for fine-tuning the LLM. Back to the purple box in the figure, we begin to recursively split `lseg(x,y)` based on these two branches, shown as the expanding arrows with "Definition 1" and "Definition 2". We noticed that the expressions after "Definition 1" expansion lead to an end of further expansions, as the predicate `lseg` itself has been eliminated. We

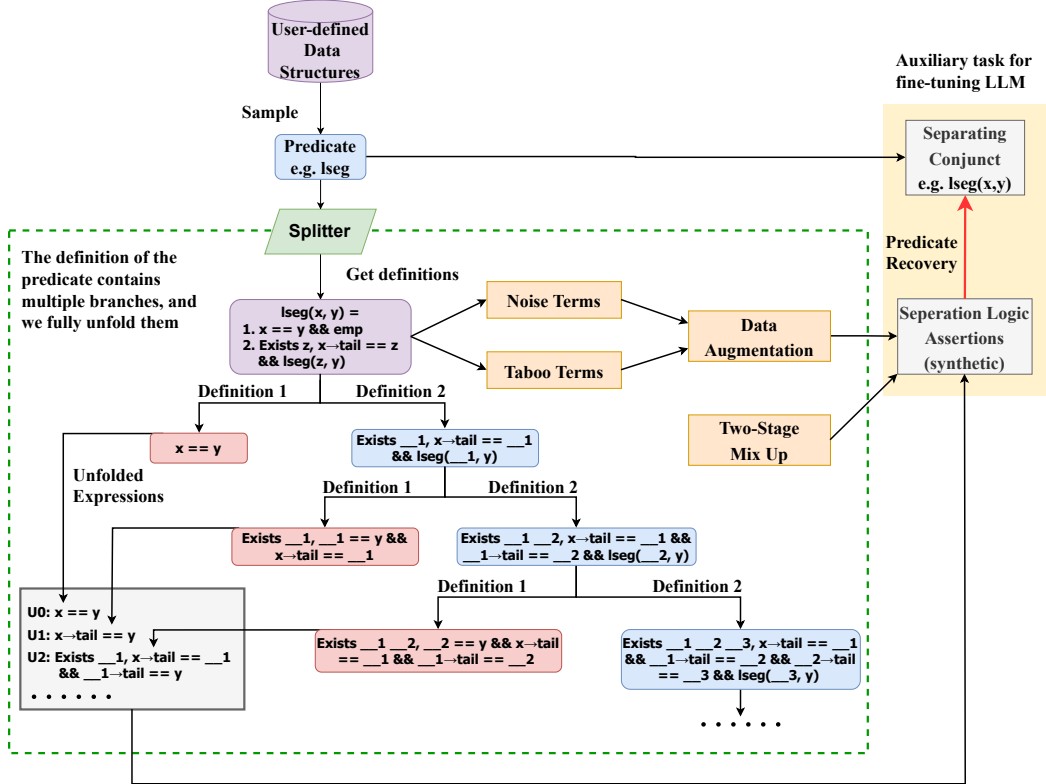

Figure 5: Our self-supervised learning paradigm for fine-tuning LLM. To solve the data scarcity issue, we design an auxiliary task called *predicate recovery*. We split the data structure completely based on its definitions, further process them from unfolded expressions to synthetic separation logic assertions, and let the LLM try to recover the original separating conjunct. Moreover, we apply multiple data augmentation and mix up strategies to refine the synthetic assertions to mimic the real ones, making the auxiliary task more challenging.

mark these expressions with red boxes and the others with blue boxes. For the expressions in blue boxes, we can further expand them based on these two branches recursively. Repeatedly, we can acquire an expansion tree as shown in the figure.

After splitting the predicate, we select the expressions in the red boxes, the terminating nodes of the expansion tree for further usage. Then, we further process them to remove the redundant temporal variables. For example, the temporal variable `__2` in the bottom red plays no significant role and can be directly replaced by program variable `y`. After the processing, we can obtain the synthetic unfolded expressions, as shown in the grey box at the left bottom corner of the figure. In this way, we successfully split the predicate into a set of unfolded expressions. We define this process as `Gen0(C)`, such as the example in the figure:

```
Gen0(lseg(x, y)) = [U0, U1, U2, ...]
```

Admittedly, there is still a gap between the generated unfolded expressions `[U0,U1,U2,...]` and the real separation logic assertions (see `S0-S5` Figure 2 or Section 4.1 for examples), where the real assertions may contain other separating conjuncts or pure proposition. To address this issue, we further employ two techniques in self-supervised learning: *data augmentation* and *mix up*. Data augmentation (53; 54) involves modifying the data by applying various transformations, such as rotation, scaling, or flipping, creating augmented versions of the original data. These altered examples offer a more diverse range of inputs for the model, helping it to generalize to unseen data. Mix up (55; 56), on the other hand, is a regularization technique that blends samples in the dataset, combining two examples by interpolating their features and labels. By averaging two data samples, mix up encourages smoother decision boundaries, mitigating overfitting and enhancing the ability

to learn from diverse data. Both data augmentation and mix up are instrumental in increasing the diversity and variability within the data, thereby boost the capacity to handle real-world scenarios. Next, we shall introduce how we adapt data augmentation and mix up in our work.

**Data augmentation.** Based on the generated unfolded expressions [U0,U1,U2,...], we try to augment them by adding more terms. Referring to Figure 5, we can see there are two right arrows from the purple box. They indicate that we derive noise terms and taboo terms based on the definition of the predicate. Taking lseg(x,y) for an example: based on its definition, we know it is an manipulation of single list and contains operations to its tail field. Therefore, the possible noise terms to add for this separating conjunct are:

```
{}=={}, {}!={}, listrep({}), {}→tail=={}, listrep(y), y→tail=={}
```

where {} could be program variables, temporal variables, or even NULL. Since the definition of lseg(x,y) already includes the memory address operation of x, more description to this memory is not allowed. Therefore, x→tail={}, listrep(x) are regarded as the taboo terms of lseg(x,y). After specifying the noise terms and taboo terms, we can randomly add more terms to the original separation expressions based on a set of certain probabilities, and we defined this process as Gen1(C) = augment(Gen0(C)), for example:

```
Gen1(lseg(x, y)) = augment(Gen0(lseg(x, y)) = augment([U0, U1, U2, ...])
= [U0 && y != 0 && ..., U1 && y→tail==0 && ..., U2 && listrep(y) && ..., ...]
```

**Mix up.** To further enhance the quality, we design a two-stage mix up strategy. Suppose we have split the predicates A, B, and obtain their unfolded expressions [A0,A1,A2,...], [B0,B1,B2,...]. We have two ways to mix up their predicates and expressions. In the first way, we mix their predicates as A*B. The corresponding expressions shall become [A0*B0,A1*B1,A2*B2,...]. It can mimic the situation that the loop invariant contains multiple predicates. In the second way, we combine their predicates A||B. The combined expressions will be [A0||B0,A1||B1,A2||B2,...]. It is very common for loop invariants to contain multiple terms separated by ||, especially for programs in the real world. Our two-stage mix up is conducted recursively with a certain probability, which results in a diverse and changeable combination of expressions. We define the aforementioned process as Gen2(C), where C is composed of one or multiple separating conjuncts.

```
Gen2(C) = Gen1(C) if C is a single conjunct,
          Gen2(A) * Gen2(B) if C = A * B,
          Gen2(A) || Gen2(B) if C = A || B
```

In the end, we can obtain a batch of synthetic separation logic assertions more close to real assertions, which can be used to create the *predicate recovery* task. In this auxiliary task, the synthetic separation logic assertions are regarded as input and the original separating conjunct is regarded as output or the label. As this whole process is fully automated, we can easily generate rich labeled data and fine-tune our LLM with this auxiliary task.

### A.2.2 Online Querying with Interactive System

Following the offline training, we obtain a LLM that is capable of inferring the separating conjuncts given the assertions. Building on this LLM and the conventional techniques of symbolic execution and entailment solver, we devise an interactive framework that can generate loop invariants online by multiple interactions. In this section, we present the loop invariant generation process for the single-layer loop case, and then demonstrate how to generalize it to the multi-layer loop case.

**Basic Interaction of Single Loop**

When focusing on single loop programs, we assume that the loop is not preceded or followed by any statements. We take the loop condition e, the loop body body and the loop precondition Pre as input, as shown in Algorithm 1. We perform a number of symbolic executions and obtain the postcondition array I for executing the loop body $0 \sim$ **MAX_NUM** times. The function true_condition computes the postcondition of the precondition when the loop condition e holds, ensuring that the loop body can be entered. We then obtain the loop invariant inferences invs from LLM using the function Infer_invs. Then, we will obtain a simplified invariant assertion inv by the function Pick_invs based on the calculated postcondition sequence I and the returned invariant sequence

**Algorithm 1:** Single_loop_inv_gen

**Data:** Pre : precondition
   e : an expression of loop condition
   body : a loop-free program fragment
**Result:** inv : loop invariant for program c = while (e) {body}

1   I[0] = Pre
2   **for** $i : 0 \rightarrow MAX\_NUM - 1$ **do**
3     |   I[i+1] = Symbolic(true_condition(I[i],e), body)
4   **end**
5   invs = Infer_invs(I)
6   inv = Pick_invs(I,invs)
7   inv_post = Symbolic(true_condition(inv,e), body)
8   **if** *entailment_solver(inv, Pre) && entailment_solver(inv, inv_post)* **then**
9     |   return inv
10   **end**
11   **else**
12     |   exit("LLM Inference Fail")
13   **end**

---

**Algorithm 2:** Infer_invs

**Data:** I: list of calculated postcondition
**Result:** inv : invariant inferred from I

1   **if** *I == []* **then**
2     |   return ""
3   **end**
4   **else**
5     |   **if** *Similar(I)* **then**
6     |     |   return Similar_extract(I)
7     |   **end**
8     |   **else**
9     |     |   sep = use_llm(model, I)
10     |     |   succ_I, fail_I = use_solver(I, sep)
11     |     |   inv_1 = Infer_invs(succ_I)
12     |     |   inv_2 = Infer_invs(fail_I)
13     |     |   inv = inv_1 || inv_2
14     |     |   return inv
15     |   **end**
16   **end**

---

`invs`. Finally, we verify the correctness of the loop invariant by `entailment_solver` via our entailment solver. If the loop invariant is correct, we return it.

The key point of the algorithm introduced above is to obtain enough information by multiple symbolic executions and output invariants according to the calculated postconditions. The former has already been mentioned in section 4.1, while the latter relies on the interaction between our LLM and traditional tools. In each round of interaction, we partition the assertion set I into two subsets `succ_I` and `fail_I` based on the separating conjunct `sep` inferred by the well-trained LLM. The subset `succ_I` consists of the assertions that are successfully replaced by `sep`, while the subset `fail_I` comprises the assertions that are not successfully replaced.

```
∀ i ∈ succ_I, i ⊢ sep * True
∀ i ∈ fail_I, i ⊬ sep * True
```

Subsequently, we apply repeated operations on the two subsets `succ_I` and `fail_I` , and eventually obtain the two invariant results `inv_1` and `inv_2`, which are combined as the solution for I. If the assertions in the set I are deemed to be similar assertions, we directly return the new assertion that is derived from the similar assertions as the solution.

**Algorithm 3:** Multi_loop_inv_gen

**Data:** Pre : precondition
e : an expression of loop condition
body : a program fragment

**Result:** inv : loop invariant for the outer loop

```
1  I[0] = Pre
2  if loop-free body then
3  │   return Single_loop_inv_gen(I[0],e,body)
4  end
5  else
6  │   (c_before, inner_e, inner_body,c_after) = Split_program(body)
7  │   for i : 0 → MAX_NUM - 1 do
8  │   │   inner_pre = Symbolic(true_condition(I[i],inner_e),c_before)
9  │   │   inner_inv = Multi_loop_inv_gen(inner_pre,inner_e,inner_body)
10 │   │   I[i+1] = Symbolic(false_condition(inner_inv,inner_e), c_after)
11 │   end
12 │   invs = Infer_invs(I)
13 │   inv = Pick_invs(I,invs)
14 │   inv_post = Symbolic(true_condition(inv,e), body)
15 │   if entailment_solver(inv, Pre)  && entailment_solver(inv, inv_post) then
16 │   │   return inv
17 │   end
18 │   else
19 │   │   exit("LLM Inference Fail")
20 │   end
21 end
```

Taking list reverse program from Section 4.1 as an example, we assume that `I=[S0,S1,S2,S3]`, and suppose that LLM returns `lseg(w,p)` in the first round of interaction. At this point, we have `succ_I = [S1',S2',S3']` and `fail_I = [S0]`.

```
S1' : v == t && p → tail == 0 && lseg(w,p) * listrep(v)
S2' : v == t && p → tail == 0 && lseg(w,p) * listrep(v)
S3' : v == t && p → tail == 0 && lseg(w,p) * listrep(v)

inv_1 :  p → tail == 0 && lseg(w,p) * listrep(v)
```

Then we perform the second round of interaction on `succ_I` and `fail_I` separately. In the recursive process of `succ_I`, we find that S1', S2' and S3' are similar assertions, so we return `inv_1`. In the recursive process of `fail_I`, we find that there is only S0 in the set, so we directly return `inv_2 : S0`. Therefore, the final answer is `p → tail == 0 && lseg(w,p) * listrep(v) || S0`.

Algorithm 2 implement the above interaction process. We denote the separating conjunct inference function by `use_llm(model,I)`, which returns the inferred separating conjunct under the given LLM `model` and the separation logic assertion set `I`. The conjunct updating function is `use_solver`, and we classify the assertions according to the results of the entailment solver. Note that we use the `Similar` and `Similar_extract` function in the termination condition judgment. The `succ_I` of the list reverse program above is a typical example. We can directly find that they are the same assertion after performing the string substitution algorithm. If Algorithm 1 fails to find a valid loop invariant, we will mask the current output of LLM and retry it.

For generating the final loop invariant, we can use the disjunction of all assertions as we did in last case, but this way may not be concise. We hope to select the smallest set of assertions that can cover all cases from them (here we use a disjunction of assertion as a set of assertions to select), so we propose a Pick Algorithm to solve this problem. Our Pick Algorithm addresses the fundamental problem of selecting a minimal set of assertions that covers all the assertions. We formulate the problem as a SAT problem for convenience. We associate each assertion with two points, namely the Cover Point that indicates the coverage of the assertion and the Pick Point that indicates the selection of the assertion. We construct relevant clauses based on assertion derivation. We compute the set

of assertions $S_i$ that can be entailed by each assertion $A_i$, that is, $A_i$ implies each element $A_j \in S_i$, where $A_i$ is included in $S_i$. We then construct a clause: $C_i \rightarrow \bigvee P_j$, indicating that any assertion in $S_i$ can cover $A_i$ if selected. Hence, our final solution is the set of assertions with $P_i$ being true under the satisfiability of $\bigwedge C_i$.

**Extension to Multi Loop Scenario**

In this section, we discuss how to extend our algorithm for single loop programs to the general multi loop case. Before, we assume that the single loop program consists of only one loop statement, and that there are no statements before or after the loop. To handle statements before or after the loop, we simply need to compute the loop body precondition from the program precondition, which is a straightforward step that we omit here. For the case of multiple loops, we focus on nested loops, which are loops that contain other loops inside their body. The case of sequential loops, which are loops that follow each other, can be easily handled by applying the algorithm for single loop programs repeatedly.

The biggest difficulty in our implementation is to find the strongest postcondition of the outer loop executing a single iteration. Since we do not know the loop invariant of the inner loop, we need to first find the loop invariant `inner_inv` of the inner loop under the current precondition `S0` of the outer loop, and this is a smaller sub-problem. Once we find `inner_inv`, we can perform symbolic execution from `S0` to get `S1`. Similarly we can find `S2`, `S3` and so on, then we can manage to generate the loop invariant of the outer loop. Finally, we only need to infer the loop invariant of the inner loop based on the loop invariant of the outer loop, and then we have completed the generation of all loop invariants of the entire program.

We present the specific algorithm implementation in Algorithm 3. The input parameters are similar to Algorithm 1, the only difference is that the loop body of Algorithm 3 can contain other loops. If `body` does not involve any other loop statements, we invoke Algorithm 1 directly. Otherwise, we will apply the `Split_program` function to decompose it into four components: the inner-pre-loop statement `c_before`, the inner loop condition `inner_e`, the inner loop body `inner_body` and the inner-post-loop statement `c_after`. We compute the pre-loop condition `inner_pre` by symbolically executing `inner_e`. By recursive calling, we can obtain the `inner_inv` for inner loop under the precondition `inner_pre`, which allows us to determine the exit condition of the inner loop. Here we use `false_condition` to computes the postcondition of the precondition when the loop condition `e` does not hold. Based on this, we can calculate the strongest postcondition for continuing to execute `c_after`, which is completing one iteration of body from `I[i]`. The remaining work is similar to Algorithm 1, and we will not go into details here.

## A.3 Examples of Programs in LIG-MM

### A.3.1 Doubly Linked List Example

```
struct list_t {
    struct list_t *prev;
    struct list_t *next;
};

/*@ Let dlistrep(l, p) = l == 0 && emp ||
      ∃ t, data_at(field_addr(l, next), t) *
              data_at(field_addr(l, prev), p) *
              dlistrep(t, l)
 */

/*@ Let dlseg(x, xp, yp, y) = x == y && xp == yp && emp ||
      ∃ z, data_at(field_addr(x, next), z) *
              data_at(field_addr(x, prev), xp) *
              dlseg(z, x, yp, y)
 */

struct list_t *iter_back(struct list_t *l, struct list_t *head)
/*@ With l_prev
```

```
    Require dlseg(head, 0, l_prev, l) * dlistrep(l, l_prev)
    Ensure  dlistrep(__return, 0)
 */
{
    struct list_t *p;
    if (l == 0) {
      return head; //@ dlseg(head,0,l_prev,0) * dlistrep(0,l_prev)
    }
    else {
      p = l; //@ l == p && dlseg(head, 0, l_prev, l) * dlistrep(l, l_prev)
      while (p != head) {
        p = p → prev;
      }
      return p;
    }
}
```

This code is changed from function list_for_each_prev of GlibC. This code traverses the entire doubly linked list along the prev pointer. Similar with singly linked list, we define `dlistrep(x,y)` to represent a doubly linked list that starts with x, where the prev of x is y (if x is not 0), `dlseg(x,xp,yp,y)` to represent a segment of doubly linked lists that starts with x and end with yp, where the prev of x is xp and the next of yp is y.

```
S0 : l == p && dlseg(head, 0, l_prev, l) * dlistrep(l, l_prev)
S1 : l != head &&
     p == l_prev && p→next == l &&
     dlseg(head,0,p→prev,p) * dlistrep(l,l_prev)
S2 : l != head && l_prev != head &&
     p→next == l_prev && l_prev → next == l && l_prev → prev == p &&
     dlseg(head,0,p→prev,p) * dlistrep(l,l_prev)
S3 : ∃ p0, l != head && l_prev != head && p0 != head &&
     p0→next == l_prev && l_prev → next == l &&
     l_prev → prev == p0 && p0 → prev == p && p → next == p0 &&
     dlseg(head,0,p→prev,p) * dlistrep(l,l_prev)

One valid loop invariant:
    l == p && dlseg(head, 0, l_prev, l) * dlistrep(l, l_prev) ||
    dlseg(head,0,p→prev,p) * dlseg(p→next,p,l_prev,l) * dlistrep(l,l_prev)
```

With symbolic execution, we can observe that p divides `dlseg(head,0,l_prev,l)` into two segments, so we can guess `dlseg(head,0,p→prev,p) * dlseg(p→next,p,l_prev,l)` and get one valid loop invariant.

### A.3.2 Singly Linked List with Multi-level Pointers Example

```
struct list {
    struct list *tail;
};

/*@ Let listrep(l) = l == 0 && emp ||
      ∃ t, data_at(field_addr(l, tail), t) * listrep(t)
 */

/*@ Let lseg(x, y) = x == y && emp ||
      ∃ z, data_at(field_addr(x, tail), z) * lseg(z, y)
 */

/*@ Let listbox_rep(x) = ∃ p, *x == p && listrep(p) */
```

```
/*@ Let listbox_seg(x,y) = x == y && emp ||
        ∃ p, *x == p && listbox_seg(&(p→tail),y)
*/

struct list ** malloc_list(void)
  /*@ Require emp
      Ensure ∃ p, *__return == p && emp
   */
  ;

void free_list(struct list * * p2)
  /*@ With p
      Require *p2 == p && emp
      Ensure emp
   */
  ;

struct list *iter(struct list *x)
  /*@ Require listrep(x)
      Ensure listrep(__return)
   */
{
  struct list * * t, * * px;
  px = malloc_list();
  t = px;
  * t = x;
  /*@ t == px && *t == x && listrep(x) */

  while ( * t != (void *) 0) {
    t = &(( *t) → tail);
  }
  x = * px;
  free_list(px);
  return x;
}
```

This code uses second-order pointers to traverse a singly linked list, and in addition to `listrep` and `lseg`, we also introduce `listbox_rep` and `listbox_seg` to represent the corresponding second-order pointer structure.

```
S0 : t == px && *t == x && listrep(x)
S1 : x != 0 &&
     *px == x && *t == x → tail && listrep( *t )
S2 : ∃ p0, x != 0 && p0 != 0 &&
     *px == x && x → tail == p0 && *t == p0 → tail && listrep( *t)
S3 : ∃ p0 p1, x != 0 && p0 != 0 && p1 != 0 &&
     *px == x && x → tail == p0 && p0 → tail == p1 &&
     *t == p1 → tail && listrep( *t)

One valid loop invariant:
     listbox_seg(px, t) * listrep( *t)
```

With symbolic execution, we can observe that the two secondary pointers, px and t, point to a segment of a singly linked list. so we can guess `listbox_seg(px,t)` and get one valid loop invariant.

### A.3.3   Tree Example

```
struct tree {
    int data;
    struct tree * left;
```

```
        struct tree * right;
        struct tree * parent;
};

/*@
  Let tree_rep(p, p_par) = p == 0 && emp ||
        ∃ p_lch p_rch,
                        data_at(field_addr(p, left), p_lch) *
                        data_at(field_addr(p, right), p_rch) *
                        data_at(field_addr(p, parent), p_par) *
                        tree_rep(p_lch, p) *
                        tree_rep(p_rch, p)
*/

/*@
  Let ptree_rep(p, p_par, p_root, p_top) = p == p_root && p_par == p_top && emp ||
        ∃ ppar_rch ppar_par ,
                        data_at(field_addr(p_par, left), p) *
                        data_at(field_addr(p_par, right), ppar_rch) *
                        data_at(field_addr(p_par, parent), ppar_par) *
                        tree_rep(ppar_rch, p_par) *
                        ptree_rep(p_par, ppar_par, p_root, p_top) ||
        ∃ ppar_lch ppar_par ,
                        data_at(field_addr(p_par, left), ppar_lch) *
                        data_at(field_addr(p_par, right), p) *
                        data_at(field_addr(p_par, parent), ppar_par) *
                        tree_rep(ppar_lch, p_par) *
                        ptree_rep(p_par, ppar_par, p_root, p_top)
*/

struct tree *Find_root(struct tree * x)
/*@ With fa root
    Require x != 0 && tree_rep(x, fa) * ptree_rep(x, fa, root, 0)
    Ensure tree_rep(__return , 0)
 */
{
    while (x → parent)
      x = x → parent;
    return x;
}
```

This code traverses the tree from x to root along the parent pointer. We define tree_rep(x,fa) to represent a tree with root x, where the parent of x is fa (if x is not 0). And we use ptree(p, p_par, p_root, p_top) to represent a part of tree with a hole at p, which means that tree_rep(p_root, p_top) = tree_rep(p, p_par) * ptree(p, p_par, p_root, p_top).

```
S0 : x != 0 && tree_rep(x, fa) * ptree_rep(x, fa, root, 0)
S1 : ∃ x0, x0 != 0 && fa != 0 &&
     x == fa && tree_rep(x0,fa) * ptree_rep(x0,fa,root,0)
S2 : ∃ x0 x1, x0 != 0 && fa != 0 && x != 0 &&
        fa → left == x0 && fa → right == x1 && fa → parent == x &&
        tree_rep(x1,fa) * ptree_rep(fa,x,root,0) * tree_rep(x0,fa) ||
     ∃ x0 x1, x0 != 0 && fa != 0 && x != 0 &&
        fa → right == x0 && fa → left == x1 && fa → parent == x &&
        tree_rep(x1,fa) * ptree_rep(fa,x,root,0) * tree_rep(x0,fa)

One valid loop invariant:
    x != 0 && tree_rep(x, fa) * ptree_rep(x, fa, root, 0) ||
    ∃ x0, x != 0 && tree_rep(x0, x) * ptree_rep(x0, x, root, 0)
```

Since S3 has 4 branches, we've omitted it here. But from S0 S1 S2 we can already find some patterns, we can guess $\exists$ `x0, x!= 0 && tree_rep(x0, x)` and get one valid loop invariant.

## A.4 Prompt Text for GPT-4

When using GPT-4, we write a common prompt text, which we add before each program of our benchmark. The prompt text is given as follows, and the examples of predicate definitions and programs can be found in the last section (A.3).

```
You will receive a program, and please fill out the 'INFILL' parts with
    suitable loop invariants.
Please only output loop invariants and no more text is needed.
You may use the defined predicates below and the data_at operation in your
    invariants.
(Note: data_at operation is an atomic operation, data_at(x, v) denotes that
    the memory location x contains the value v.)
Definitions:
1. xxx
2. xxx
xxx
Program:
xxx
```

## A.5 Licenses of Existing Benchmarks Used in LIG-MM

In the components of our LIG-MM, the programs derived from course homework are collected on our own, and the other programs are selected from existing benchmarks. Here, we will list the sources and licenses of the existing benchmarks used in our work.

- SLING ([18]; [19]). The official website of SLING is https://github.com/guolong-zheng/sling, and we used the PLDI version of their benchmark listed in the repository. Currently, there is no license on their website. In our work, we strictly follow their instructions, and we believe there is no risk of infringement.

- SV-COMP ([28]). The official website of SV-COMP is https://gitlab.com/sosy-lab/benchmarking/sv-benchmarks/-/tree/main/c, and it follows the Apache-2.0 license. The SPDX-FileCopyrightText is 2011-2013 Alexander von Rhein, University of Passau and 2011-2021 The SV-Benchmarks Community.

- Linux Kernel ([29]). We use the programs from https://github.com/torvalds/linux/, which collects the code of the Linux kernel. There are multiple licenses in this repository and the programs we select all follow the GPL-2.0 license (https://github.com/torvalds/linux/blob/master/LICENSES/preferred/GPL-2.0).

- GlibC ([30]). The official GitHub repository of GlibC is https://github.com/kraj/glibc/blob/master/. The license of their code is GNU LESSER GENERAL PUBLIC LICENSE (LGPL) v2.1 https://github.com/kraj/glibc/tree/master?tab=LGPL-2.1-2-ov-file.

- LiteOS ([31]). The official website of LiteOS is https://gitee.com/openharmony/kernel_liteos_m and their license is BSD 3-Clause License https://gitee.com/openharmony/kernel_liteos_m/blob/master/LICENSE.

- Zephyr ([32]). The official GitHub repository of Zephyr is https://github.com/zephyrproject-rtos/zephyr, and it follows the Apache-2.0 license (https://github.com/zephyrproject-rtos/zephyr/blob/main/LICENSE).

## A.6 Limitation, Impact, and Outlook

While our work represents a significant advancement in the field of loop invariant generation, it is not without its limitations. Firstly, although our proposed LLM-SE framework demonstrates strong performance on various benchmarks, its pass rate is not yet sufficient for seamless integration

into all real-world software applications. The reliance on the accuracy and comprehensiveness of separation logic predicates means that any gaps in these definitions can potentially limit the model's effectiveness. Moreover, while our method is designed to generalize across various data structures and multi-loop scenarios, specific edge cases and complex data manipulations remain that may not be fully addressed. The computational overhead associated with symbolic execution and the interactive querying process may also pose challenges for scaling up to very large and complex software systems.

As for the social impacts. On the positive side, enhancing the ability to verify and validate software automatically can lead to more reliable and secure systems. This is particularly crucial in domains such as healthcare, finance, and automotive industries, where software correctness can directly and significantly impact safety and security. By reducing the need for extensive manual intervention in the verification process, our approach can also democratize access to high-assurance software development, enabling smaller teams and organizations to build reliable systems without requiring deep expertise in formal methods. However, there are also potential negative implications to consider. The automation of verification tasks may lead to job displacement for some roles traditionally involved in software testing and verification. Additionally, as with any AI-driven technology, there is a risk of over-reliance on automated tools, which may result in a false sense of security if the tools are not used properly or if their limitations are not fully understood. Ensuring transparency in how these tools work and maintaining human oversight in critical decision-making processes will be essential to mitigate these risks.

Looking ahead, several promising directions for future research and development exist. One key area is enhancing our LLM-SE framework to further improve its accuracy and applicability. This could involve refining the self-supervised learning paradigm, exploring more sophisticated symbolic execution techniques, and developing more comprehensive sets of separation logic predicates. Additionally, integrating our approach with other program analysis tools and methodologies could provide a more holistic solution for software verification.

Expanding the scope of our benchmarks to include an even wider range of real-world software systems will also be important for validating and improving the robustness of our methods. Collaboration with industry partners could facilitate access to more diverse datasets and real-world use cases, driving further innovation and practical impact.

Finally, fostering a community of researchers and practitioners around loop invariant generation and related areas will be crucial for sustaining progress. By sharing our findings, tools, and datasets openly, we aim to contribute to the collective knowledge and capabilities of the field, encouraging further exploration and refinement of automated software verification techniques.

In summary, while our work marks a significant step forward, it also opens up numerous opportunities for future research and development. By addressing the current limitations, understanding the broader social impacts, and continuing to innovate, we can move closer to realizing the full potential of automated loop invariant generation in creating reliable, secure, and high-assurance software systems.

