# Towards General Loop Invariant Generation: A Benchmark of Programs with Memory Manipulation

## 1 Overview of Supplementary Material

Due to the page limit in the submitted paper, we shall provide more detailed information on our proposed benchmark dataset LIG-MM and the our proposed framework LLM-SE. The supplementary material is organized as follows:

- Sec. 2: Dataset Accessibility and Documentation.
- Sec. 3: Program Example in LIG-MM.
- Sec. 4: Licenses.
- Sec. 5: Details of LLM-SE Framework.
- Sec. 6: Prompt Design for GPT
- Sec. 7: Related Work.
- Sec. 8: Discussion of Limitation, Social Impact, and Outlook.

## 2 Dataset Accessibility and Documentation

**Dataset Documentation:** We have documented our dataset for intended researchers as required. The website of our benchmark dataset is available at the following link: `https://anonymous.4open.science/r/LIG-MM-NeurIPS24/`, which includes the programs collected from various sources, the format detail of examples and the code to reproduce the results in our experiments. The link to download the models after fine-tuning is `https://mega.nz/file/M9FEWCjD#QkAQLu7UERPk4Xgb-Rer4U7lfKy7P3rdQeY_p-b8nhM`.

**Dataset Statistics:** As we mentioned in our paper, the benchmark programs in existing papers mostly contain numerical programs. To fill the lack of benchmarks for general loop invariant generation, we propose LIG-MM, a loop invariant generation benchmark of memory manipulation programs. Table 1 below shows the basics of the code in LIG-MM. Our programs come from four main sources: course codes, competition codes, previous relevant work, and the actual system codes. The programs are modified into a unified format for better usage. Multiple examples are shown in Sec. 3, and the licenses of benchmarks can also be found in Sec. 4.

- *Course codes.* The course code is mainly derived from homework programs on the data structure course and programming language course. The detailed course number and college name are covered due to the anonymity of this paper. These programs contain the most diverse data structures and multi-level pointer operations among the sources of our benchmark.

Submitted to the 38th Conference on Neural Information Processing Systems (NeurIPS 2024) Track on Datasets and Benchmarks. Do not distribute.

Table 1: Statistics of our proposed LIG-MM benchmark.

|  | Concrete Resources | Number of Programs | Data Structure Types |
|---|---|---|---|
| Course codes | Course homework programs | 187 | sll, dll, tree, hash-table |
| Competition codes | SV-COMP [1, 2] | 27 | sll, dll, tree, hash-table |
| Previous benchmark | SLING [3, 4] | 15 | sll, dll, tree |
| Real-world programs | Linux Kernel [6] | 23 | sll, dll, hash-table |
| Real-world programs | GlibC [5] | 13 | dll, hash-table |
| Real-world programs | LiteOS [7] | 12 | dll |
| Real-world programs | Zephyr [8] | 35 | sll, dll, hash-table |
| Overall | - | 312 | sll, dll, tree, hash-table |

- *Competition codes.* SV-COMP[1, 2] is a competition on software verification, which provides a benchmark for verification tasks for comparing verification tools. Originating from competition, this dataset encompasses various verification tasks, providing a comprehensive set of real-world and synthetic examples for testing the effectiveness and efficiency of verification techniques. In our LIG-MM, we select programs from the 2022 competition benchmark.

- *Previous relevant work.* SLING [3, 4] uses traditional dynamic analysis techniques to generate invariants. Other than loop invariants, SLING can also generate preconditions and post-conditions for function. Therefore, not all their benchmarks include the inference of loop invariant or even contain a loop (they use function calls to replace loops). After selection, we choose some of the programs in its benchmark and turning them into a uniform code style.

- *Real-world system codes.* To make the data in LIG-MM closer to real-world software environments, we decide to select more programs from several well-known software and systems. Among them, GlibC [5] is the GNU implementation of the C standard library, providing essential functionalities for numerous applications. Additionally, we have incorporated programs from the Linux Kernel [6], a widely used and highly-regarded operating system kernel that serves as the foundation for countless devices and systems worldwide. To further enhance the diversity of our dataset, we have included LiteOS [7], a lightweight operating system designed for IoT devices, and Zephyr [8], another versatile operating system known for its applicability in resource-constrained environments.

By integrating these varied sources, LIG-MM captures a broad spectrum of programming practices and challenges and ensures that our benchmark is robust and representative of the complexities encountered in multiple scenarios, such as real-world software development and verification. Unlike the numerical program benchmark in previous works [9, 10, 11, 12, 13, 14], our benchmark does not contain pure numerical procedures, all of our programs are related to at least one certain data structure. The data structures we have selected include singly linked lists(sll), doubly linked lists(dll), trees, and hash-tables. In addition, our benchmark includes the usage of multi-level pointers and various pointer arithmetic.

**Long-term Preservation and Maintenance:** To ensure the longevity and relevance of our proposed LIG-MM benchmark, we will maintain a dedicated public repository on GitHub, facilitating easy access and version control. Regular updates will be made to incorporate new programs, improvements, and community contributions. We encourage open-source dataset, benchmark and codes, inviting researchers to contribute and review submissions to ensure quality. Comprehensive documentation will guide users on structure, usage, and contributions. Engaging with the research community through workshops and conferences will help gather feedback for continuous enhancement. Through these measures, we aim to provide a sustainable resource for ongoing advancements in program verification, loop invariant generation, and memory manipulation analysis.

**Terms of Use and License:** We have chosen the GPL-2.0 license for our benchmark dataset, and the detailed license is clearly stated on our dataset website.

**Discussion of Personally Identifiable Information**. As we mentioned before, the course code of our benchmark is mainly derived from homework programs on the data structure course and programming language course. The detailed course number and college name are covered to avoid the link of privacy. Thus, we can confirm that our LIG-MM benchmark does not contain personally identifiable information or offensive content.

## 3 Program Example in LIG-MM