# OpenReview forum: "Towards General Loop Invariant Generation: A Benchmark of Programs with Memory Manipulation"
_NeurIPS.cc/2024/Datasets_and_Benchmarks_Track — NeurIPS 2024 Track Datasets and Benchmarks Poster_

### Official Review · Reviewer_EQmi · 2024-07-23
**Review Comments**

**Rating:** 5
**Confidence:** 4
**Clarity:** The paper is well written.

**Review:**

This submission presents a significant advancement in the field of program verification, particularly in automating the generation of loop invariants for complex software systems. The novel approach and comprehensive evaluation make this a noteworthy contribution, despite some potential challenges in practical application and generalization.

**Strengths:**

1. Creation of the LIG-MM benchmark: This benchmark addresses the challenges in generating loop invariants for programs that deal with complex data structures and memory manipulation, a notable gap in existing benchmarks.
2. Evaluating multiple baselines including GPT-4 on LIG-MM benchmark: The authors evaluate multiple baselines on LIG-MM to analyze the capacity of existing approaches to generate loop invariants in LIG-MM benchmark.
3. Introduction of the LLM-SE framework: This framework integrates LLMs with symbolic execution, fine-tuned via self-supervised learning, to automate the generation of loop invariants.

**Additional Feedback:**

This work provides a benchmark, LIG-MM, for the field of program verification to analyze the capacity of existing approaches to generate loop invariants. However, it is lacking in experimental analysis and does not provide good insights to the reader on how to promote the development of the field.

**Correctness:**

The benchmark creation, testing, and evaluation methodology are all sound. This paper appears correct about its claims.

**Documentation:**

The authors did not provide a URL for reviewer access to the dataset.

**Ethics:**

No ethical concerns.

**Limitations:**

Yes, the authors have included a comprehensive discussion on limitations.

**Opportunities For Improvement:**

1. LIG-MM includes a variety of data structures, such as linked lists, trees, hash-tables, and various user-defined data structures. However, this paper lacks statistics on the number of samples for different data structures. Could there be an uneven distribution of categories?
2. The paper lacks a fine-grained experimental analysis of the current methods on LIG-MM. Do different methods perform differently on various data structures? What are the main weaknesses of the current methods? If the authors could provide an in-depth discussion and analysis of the shortcomings of the current methods, it would offer more insights to the readers and better promote the development of this field.
3. The paper lacks ablation experiments for LLM-SE. How much does the performance improve when the authors’ proposed framework LLM-SE is added to the baseline (CodeGen)? This result would better help readers understand the importance of the training framework proposed by the authors.

**Relation To Prior Work:**

The paper clearly discusses how it relates to prior work.

**Summary And Contributions:**

This paper introduces the LIG-MM benchmark for loop invariant generation, targeting programs that manipulate complex data structures and memory. For this benchmark, the authors propose a new approach, LLM-SE, that integrates LLMs with symbolic execution and is fine-tuned via self-supervised learning to automate loop invariant generation. However, the paper does not provide statistics on the number of samples for different data structures, raising concerns about potential uneven distribution. Additionally, there is no in-depth analysis of current methods on LIG-MM, nor does it include ablation experiments to show the performance improvement when the LLM-SE framework is added to the baseline (CodeGen).

---

> ### Author Rebuttal · Authors · 2024-08-15
>
> Thank you for your valuable comments. We try to answer the questions one by one and hope we can alleviate your concerns.
>
> ***Q1: Lacks statistics on the number of samples for different data structures.***
> About the statastic details of our benchmark, it is indeed our negligence, here we will provide more detailed analysis:
>
> |Data structure|Count|Avg. program lines|Avg. annotation lines|Avg. number of predicates in invariant|
> |-|-|-|-|-|
> |Data-agnostic singly linked list|56|40|12|5|
> |Data-bearing singly linked list|40|42|14|6|
> |Data-agnostic doubly linked list|85|50|15|6|
> |Data-bearing doubly linked list|48|55|17|7|
> |Data-agnostic listbox|40|62|15|5|
> |Data-bearing listbox|20|64|17|6|
> |Hashtable|6|80|30|8|
> |Tree|10|83|25|18|
> |Mixed of sll and dll|7|45|9|5|
> |Total|312|52|15|6|
>
> Moreover, the programs collected in our benchmark are the most diverse and close to real world, please refer to Table 1 of the submitted PDF or the table in our reply to Reviewer rGBC for more details.
>
> ***Q2: Do different methods perform differently on various data structures? What are the main weaknesses of the current methods?***
>
>  Thank you for your valuable suggestion. In our submitted paper (Fig. 4), we have shown different methods' pass rates on every source of programs in our LIG-MM. During our rebuttal, we analyzed the different methods' performance on different data structures, as the pass rate @ 8 is shown in the table below:
>
> |Method|SLING|AutoSpec|GPT4|LIG-SE (ours)
> |-|-|-|-|-|
> |Single linked list|33.33%|0.00%|34.38%|61.46%|
> |Double linked list|24.81%|0.00%|13.53%|32.33%|
> |Listbox|0.00%|0.00%|6.67%|41.67%|
> |Others|13.04%|0.00%|8.70%|30.43%|
> |Total|21.79%|0.00%|17.68%|42.95%|
>
> As we can see, our proposed method LIG-SE has shown considerable performances on all kinds of data structures. For the baselines, GPT4 performs well on the single linked list while SLING performs well on the double linked list, but both methods fall short when applied to more complex or less common structures like Listbox, where LIG-SE excels with a 41.67% pass rate. This indicates that while baseline methods like SLING and GPT-4 can handle simpler structures to some extent, they struggle with more intricate memory manipulations.
>
> The overall performance of LIG-SE, achieving a total pass rate of 42.95%, highlights its robustness and flexibility. In contrast, the inconsistent results from SLING and GPT-4, particularly in handling specific data structures beyond single and double linked lists, reveal the limitations of these methods. AutoSpec, with a near-zero performance across all categories, underscores the need for significant improvement in handling diverse data structures within program verification.
>
> In summary, while existing methods show varying success with specific data structures, their overall performance is inconsistent and insufficient for comprehensive program verification. LIG-SE fills this gap by offering a more reliable and effective solution for handling a wide range of data structures.
>
> ***Q3: How much does the performance improve when the authors’ proposed framework LLM-SE is added to the baseline (CodeGen)?***
>
> Thanks for your kind reminder, it is indeed our negligence to miss the ablation experiments. We are committed to incorporating more extensive ablation studies in the revised paper. As shown in the table below, preliminary results indicate intuitive findings:
>
> |Method|SLING|Auto-Spec|GPT4|LIG-SE (ours) |w.o. data aug.|w.o. SSL|w.o. SE (original CodeGen)|
> |-|-|-|-|-|-|-|-|
> |pass rate @1|12.82%|0.00%|12.50%|38.78%|27.88%|8.33%|0.00%|
> |pass rate @8|21.79%|0.00%|17.68%|42.95%|32.05%|13.46%|0.00%|
>
> We conducted an ablation study by sequentially removing key components from our LIG-SE model. First, we removed the data augmentation module (data aug.), followed by the self-supervised training module (SSL), and finally the interaction with the symbolic execution module (SE), which effectively reduced the model to the original large language model, CodeGen.
>
> The results show that the influence of the self-supervised learning component is critical—our LIG-SE struggles to function effectively without it. In contrast, the data augmentation module plays a more supplementary role, acting as the icing on the cake. As expected, the direct deployment of CodeGen without these enhancements did not perform well.
>
> ***Q4: The authors did not provide a URL for reviewer access to the dataset.***
>
> Sorry for the confusion. Actually, we have provided a URL[1] in the supplementary material. Please refer to [1], which includes our benchmark programs collected from various sources, the format detail of examples and the code to reproduce the results in our experiments. We also provide a link[2] to download the models after fine-tuning.
>
> [1] https://anonymous.4open.science/r/LIG-MM-NeurIPS24
>
> [2] https://mega.nz/file/M9FEWCjD#QkAQLu7UERPk4Xgb-Rer4U7lfKy7P3rdQeY_p-b8nhM

---

> > ### Author Response · Authors · 2024-08-30
> >
> > Dear reviewer EQmi,
> >
> > Thank you very much for your time and effort in reviewing our work. As the rebuttal deadline is approaching, we would sincerely appreciate it if you could review our responses to ensure they adequately address your concerns. Please let us know if any remaining issues or further clarifications are needed. We are committed to improving our work and are eager to address any additional points you may have.
> >
> > Best regards,
> >
> > The authors

---

### Official Review · Reviewer_4YxC · 2024-07-24
**Review 860**

**Rating:** 7
**Confidence:** 3
**Correctness:** The dataset is constructed in a sound…
**Clarity:** The paper is well-written and easy to…

**Review:**

The paper proposes a new benchmark that can benefit the community and a new inspiring method. It also includes sufficient background and related works, making it easy to follow. However, the paper needs some revisions to include more details on the benchmark and method.

**Strengths:**

- The paper includes sufficient related works and background, which is easy to understand and follow.

- The proposed interactive framework that incorporates LLM and symbolic execution is reasonable and effective. The unfolded predicate expressions have some obvious patterns that the LLMs should be able to capture. The symbolic execution can generate such expressions for LLM to predict the loop invariants. Experimental results also verify its effectiveness.

**Additional Feedback:**

The authors can include more details, human assessment, and experiments on the benchmark and method.

**Documentation:**

The data release the code and data. It gives details of the data collection but the data organization is not given yet.

**Ethics:**

The paper has no ethical concerns.

**Limitations:**

The authors adequately addressed the limitations

**Opportunities For Improvement:**

- As a benchmark paper, it does not include enough details, analysis, and assessment of the data. For example, structure distribution, program length, or properties of the groundtruth(reference) invariants. The authors need to demonstrate the difference between the proposed and previous benchmarks.

- The paper also lacks details of training data. For example, the number of defined predicates, the number of unfolded expressions, and the success rate with regard to the number of training samples. The predicate number may reflect how the proposed method generalizes to other data structures and other types of programs.

- More method's experimental results are needed. Most importantly, the method's performance on previous benchmarks such as CODE2INV, and ablation studies by excluding different data augmentation methods.

**Relation To Prior Work:**

The paper includes sufficient related work and discusses the difference.

**Summary And Contributions:**

The paper proposes a new loop invariant generation benchmark focusing on memory manipulation programs. The authors collect programs with diverse data structures from courses, competitions, previous works, and real systems. The paper also proposes a new framework to train and incorporate LLM with symbolic execution for loop invariant generation. It unfolds the predicates specific for four data structures: singly linked lists, doubly linked lists, trees, and hash-tabless, and trains the LLM to output predicates given unfolded expression. Experimental results show the proposed method achieves significant improvement on the proposed benchmark.

---

> ### Author Rebuttal · Authors · 2024-08-15
>
> We appreciate the reviewer for the valuable comments. We set out below our responses to each of your questions.
>
> ***Q1: Lack of enough details of the data and the difference between the proposed and previous benchmarks.***
>
> Sorry for the inconvenience. In our work, we provide a loop invariant generation benchmark for programs with memory manipulations. In contrast, existing benchmarks focus on numerical programs that do not contain any data structures. **To this end, our benchmark is fundamentally different from existing works.** Moreover, the programs collected in our benchmark are the most diverse and close to real world, please refer to Table 1 of the submitted PDF or the table in our reply to Reviewer rGBC for more details.
>
> About the statastic details of our benchmark, it is indeed our negligence, here we will provide more detailed analysis:
>
> |Data structure|Count|Avg. program lines|Avg. annotation lines|Avg. number of predicates in invariant|
> |-|-|-|-|-|
> |Data-agnostic singly linked list|56|40|12|5|
> |Data-bearing singly linked list|40|42|14|6|
> |Data-agnostic doubly linked list|85|50|15|6|
> |Data-bearing doubly linked list|48|55|17|7|
> |Data-agnostic listbox|40|62|15|5|
> |Data-bearing listbox|20|64|17|6|
> |Hashtable|6|80|30|8|
> |Tree|10|83|25|18|
> |Mixed of sll and dll|7|45|9|5|
> |Total|312|52|15|6|
>
> ***Q2: Lacks details of training data.***
>
> Sorry for the confusion. The number of data used for offline training is 200,000, with all kinds of predicate mix-ups. Notably, the predicates used in offline training are only in their standard formulation, while the definitions of predicates in the testing benchmark include multiple variants, such as different field names, different numbers of fields, and different usage of data/link fields. Therefore, we consider our benchmark to be quite challenging in generalization.
>
> ***Q3: The method's performance on CODE2INV.***
>
> We apologize for the lack of explanation regarding the distinction between numerical programs and memory-related programs. These two categories of programs have fundamental differences that influence the approach and tools used for verification. We consider it unsuitable to test our method in numerical program verification benchmarks (like CODE2INV).
> - The core difference lies in the nature of loop invariants that need to be verified. The loop invariants in numerical programs typically involve computing a function $\mathcal{f}$ that relates all variables and the loop counter $\mathcal{i}$. In contrast, memory-related program verification focuses on understanding the shape and structure of memory as it changes over execution.
> - Due to these differences, traditional verification methods[1] often reflect this distinction by focusing on one type of program over the other. For example, in SLING[2], the benchmark exclusively contains memory-related programs, as it is tailored to address the specific challenges associated with verifying memory structures. Conversely, numerical program verification tools like those evaluated in CODE2INV focus exclusively on numerical programs.
> - The loop invariants in numerical and memory-related programs also take on different forms. In numerical programs, invariants often take the form of equations or inequalities involving numeric variables. For memory-related programs, the invariants describe properties of data structures, such as whether a linked list is acyclic or whether a binary tree is balanced. Several studies have explored these distinctions, highlighting the specialized nature of verification tools in each domain[3][4].
>
> Given the existing focus on numerical programs in existing papers, there is a notable gap in the benchmarks for memory-related program verification. The primary purpose of our work is to address this gap by providing a memory-related loop invariant generation benchmark. This benchmark is designed to evaluate and enhance tools specifically aimed at memory-related verification, complementing existing numerical benchmarks.
>
> ***Q4: Ablation studies by excluding different components.***
>
> Thanks for your suggestion. We are committed to incorporating more extensive ablation studies in the revised paper. Preliminary results indicate intuitive findings:
>
> |Method|SLING|Auto-Spec|GPT4|LIG-SE (ours) |w.o. data aug.|w.o. SSL|w.o. SE (CodeGen)|
> |-|-|-|-|-|-|-|-|
> |pass rate @1|12.82%|0.00%|12.50%|38.78%|27.88%|8.33%|0.00%|
> |pass rate @8|21.79%|0.00%|17.68%|42.95%|32.05%|13.46%|0.00%|
>
> We conducted ablation study by sequentially removing key components from LIG-SE. First, we removed the data augmentation module (data aug.), followed by the self-supervised training module (SSL), and finally the interaction with the symbolic execution module (SE), which effectively reduced the model to the original large language model, CodeGen.
>
> The results show that the influence of the self-supervised learning component is critical—LIG-SE struggles to function effectively without it. In contrast, the data augmentation module plays a more supplementary role. As expected, the direct deployment of CodeGen without these enhancements did not perform well.
>
> ***Q5: The data organization is not given yet.***
>
> Sorry for the inconvenience. We recognize the importance of clear data structure and organization for ease of use and reproducibility. To address this, we have updated our data release documentation in our anonymous repository to include a comprehensive explanation of how the data is organized, and place one Readme in each sub-folder. It covers the structure of the files and directories and the categorization of different data sets. Our goal is to ensure that all users can easily navigate and utilize the data in their own research.
>
> [1] Efficient summary reuse for software regression verification.
>
> [2] SLING: using dynamic analysis to infer program invariants in separation logic.
>
> [3] Loop invariants: Analysis, classification, and examples.
>
> [4] Loop invariant synthesis in a combined abstract domain.

---

> > ### Comment · Reviewer_4YxC · 2024-08-28
> >
> > Thanks for the responses. The new experiments and discussion of the difference from previous benchmarks have addressed my concerns. I have raised my score.

---

> > > ### Author Response · Authors · 2024-08-28
> > >
> > > Thank you! We will continue to polish our paper, and thanks again for your valuable comments.

---

### Official Review · Reviewer_HX1p · 2024-07-25
**Review of Submission 860**

**Rating:** 9
**Confidence:** 3
**Correctness:** The claims appears to be correct.
**Clarity:** The paper is well-written.

**Review:**

This is very impressive work.
The authors successfully identify a shortcoming in the ML + loop invariant literature and propose a highly effective method grounded in separation logic and implemented with symbolic execution.
The benchmark itself is already a solid contribution, and the integration of symbolic execution and separation logic makes this all the more impressive.
This work should be well-received by the ML + formal verification community.

**Strengths:**

The authors identify and study an important problem that is not covered by existing ML + formal methods works.
The paper is well-written, the benchmark is useful, and the integration of LLM + symbolic execution is impressive.
Overall, this is a very solid paper.

**Additional Feedback:**

N/A

**Documentation:**

The code is available and documented.

**Ethics:**

There are no ethical concerns.

**Limitations:**

The authors sufficiently address the limitations.

**Opportunities For Improvement:**

In Section 2, the technical presentations of separation logic and symbolic execution are dense.
Additional figures would be helpful to readers not familiar with either topics, and some Section 5 text could be moved to the Appendix if space is needed.

**Relation To Prior Work:**

Relation to prior work is sufficiently discussed.

**Summary And Contributions:**

This paper proposes a benchmark for formal verification, wherein the objective is to generate loop invariants for programs that manipulate memory (e.g., linked lists).
This addresses a need for "realistic" datasets, as many existing loop invariant benchmarks do not contain memory manipulations.
Because many existing methods, particularly LLMs, fail to find such memory-dependent loop invariants, the authors propose a framework that interleaves symbolic execution with LLMs to great effect.
The symbolic execution + LLM approach is shown to be effective, and the code is made available to the reviewers.

---

> ### Author Rebuttal · Authors · 2024-08-15
>
> Thank you for your review and acknowledgment of our work. Here are our specific responses to refine our paper based on your comments.
>
> ***Q1: The layout of the paper needs to be optimized.***
>
> Thanks for your kind reminder, and we apologize for the inconvenience in reading our paper. Since we can not edit the original paper PDF during rebuttal due to the policy of NeurIPS, we shall update our paper in the camera ready version. We plan to move the description of settings in Section 5 to the Appendix, and leave more space to introduce symbolic execution and separation logic.
>
> For now, we recommend these symbolic execution and separation logic materials[1][2][3][4][5] for readers not familiar with this field. We apologize again for our oversight.
>
> [1] Berdine J, Calcagno C, O’hearn P W. Symbolic execution with separation logic[C]//Programming Languages and Systems: Third Asian Symposium, APLAS 2005, Tsukuba, Japan, November 2-5, 2005. Proceedings 3. Springer Berlin Heidelberg, 2005: 52-68.
>
> [2] https://en.wikipedia.org/wiki/Separation_logic
>
> [3] https://www.code-intelligence.com/blog/using-symbolic-execution-fuzzing
>
> [4] https://softwarefoundations.cis.upenn.edu/slf-current/index.html
>
> [5] https://softwarefoundations.cis.upenn.edu/vc-current/index.html

---

### Official Review · Reviewer_rGBC · 2024-07-26

**Rating:** 4
**Confidence:** 5
**Clarity:** The paper is relatively well written.

**Review:**

Loop invariant inference is a foundational challenge in software verification. It is important to have a comprehensive benchmark suite to address this challenge. This paper has a strong motivation, however, the collected programs are quite limited because only a small fraction (i.e., 83 out of 312) is from real-world system code.

**Strengths:**

- This paper studies a core problem in software verification -- loop invariant inference.  Programs with memory manipulations are common in real-world systems but rarely included in existing benchmarks.
- Targeting real-world systems like Linux kernel, GlibC, LiteOS, and Zephyr is novel and important in the long run.
- Large language models integrated with symbolic execution traces outperform previous approaches.

**Additional Feedback:**

What prevents the authors from collecting a large set of programs from real-world systems? How current ones are selected?

**Correctness:**

Several programs are collected from real-world system programs, but the selection criteria are unclear.

**Documentation:**

Large systems like Linux kernel and GlibC have hundreds of thousands or millions of lines of code. It is unclear how a tiny and specific fraction (i.e., around 20) of programs are selected.

**Limitations:**

There is no discussion about potential limitations.

**Opportunities For Improvement:**

There is a huge potential for transforming large systems like the Linux kernel into loop-invariant benchmarks. The idea is novel and visionary; however, the execution is barely minimal. Instead of taking a tiny fraction from four different real-world systems, it would be way more convincing even if only one particular real-world system is used for collecting programs but with hundreds or thousands more programs.

**Relation To Prior Work:**

Yes, the key difference is a set of programs with memory manipulations, while the previous benchmarks usually do not involve memory manipulation.

**Summary And Contributions:**

This paper proposes a small benchmark of programs that involves memory manipulations for loop invariant inference, of which 83 programs are newly collected from real-world systems (e.g., Linux kernel, Glibc, LiteOS, Zephyr).  Furthermore, this paper shows that, with symbolic execution traces, large language models can help to find correct loop invariants.

---

> ### Author Rebuttal · Authors · 2024-08-15
>
> Thank you for your time, thorough comments, and kind suggestions. We shall clarify our benchmark settings and refine our paper as suggested.
>
> ***Q1: The selected programs of real-world system is only a small fraction, and the selection criterias are unclear.***
>
> Thanks for your kind reminder and we apologize for the missing explanation. We acknowledge that the number of real-world systems in our benchmark is limited, but this is an inherent challenge:
>  - Firstly, it is essential to note that our focus was on extracting and analyzing memory manipulations related to data structures within these systems. However, the number of meaningful data structure operations within the same system is typically limited, as these operations often exhibit similarity due to the consistent definition and role of data structures within a given system. For instance, in LiteOS, various modules define different data structures, like the C struct for storing task, the C struct for semephore, etc(some of the definition can be seen below). These data structure are linked in different linked list for different purpose, but there implementation are all based on the polymorphic doubly linked list LOS_DL_LIST. For our benchmarking purposes, these repetitive code patterns do not provide significant meaning. Therefore, we selected the relevant operation functions based on the LOS_DL_LIST definition for testing.
> ```
> typedef struct LOS_DL_LIST {
>     struct LOS_DL_LIST *pstPrev;
>     struct LOS_DL_LIST *pstNext;
> } LOS_DL_LIST;
>
> struct LosSemCB{
>     UINT16 semStat;
>     UINT16 semCount;
>     UINT16 maxSemCount;
>     UINT16 semID;
>     struct LOS_DL_LIST semList; /**< Queue of tasks that are waiting on a semaphore */
> } ;
> typedef struct tagEvent {
>     UINT32 uwEventID;
>     LOS_DL_LIST stEventList; /**< Event control block linked list */
> } EVENT_CB_S;
> typedef struct {
>     LOS_DL_LIST sortLinkNode;
>     UINT64      responseTime;
> } SortLinkList;
> struct LosTaskCB{
>     ... /** Some Field */
>     struct LOS_DL_LIST pendList;
>     struct LOS_DL_LIST timerList;
>     struct LOS_DL_LIST joinList;
>     EVENT_CB_S event;
>     UINT32 eventMask;
>     UINT32 eventMode;
>     void *msg;
>     INT32 errorNo;
> };
> struct LosSemCB{
>     UINT16 semStat;
>     UINT16 semCount;
>     UINT16 maxSemCount;
>     UINT16 semID;
>     struct LOS_DL_LIST semList; /**< Queue of tasks that are waiting on a semaphore */
> } ;
> ```
>
>  - On the other hand, selecting code samples from different systems, each with unique data structure definitions and operations, allows for a more comprehensive evaluation of the tool's effectiveness. This approach enables us to better assess and demonstrate the tool's capabilities across a diverse range of code scenarios.
>  - Furthermore, although only 83 of the 312 programs in our dataset are real-world systems, the remaining programs are still meaningful. Other programs include previous benchmarks, competition codes, and course homework programs. Performing program verification and invariant generation on these types of programs is also valuable, as they contribute to the overall robustness and versatility of full evaluation.
>  - Moreover, in our benchmark, we rely on the iteraction with traditional annotation-based program verification tools. For each input program, necessary annotations must be manually provided, making it challenging to collect a large number programs. Manual annotations are common in the program verification field, as supported by existing literature[1][2][3][4] (including the AutoSpec[4] baseline in our experiments).
>
> We would also like to highlight that existing benchmarks in this area are not mandatorily required to be large in scale. In both program verification[4][5][6][7] and formal verification[8][9][10][11] fields, a collection of about one hundred instances is considered substantial. In the table below, we compared the number of benchmark programs in our work with other relevant works, including the baselines[4][5] in our experiments and the foundamental work Code2Inv[6]. The scale of our dataset is in line with these established benchmarks, reflecting the depth and quality of analysis rather than the sheer quantity of data.
>
>  We recognize the importance of expanding our dataset and are committed to further enhancing its comprehensiveness. In future work, we plan to invest additional resources and effort to supplement our dataset with more real-world programs.
> Paper|Benchmark Program Count|
> |-|-|
> |AutoSpec[4]|257 (6 from real-world software system)|
> |SLING[5]|157 (4 from real-world software system)|
> |Code2Inv[6]|133 (0 from real-world software system)|
> |CLN2INV[7]|124 (0 from real-world software system)|
> |**Our Work**|312 (83 from real-world software systems, others from students' daily course codes and previous competitions/benchmarks)|
>
> [1] Verifast: A powerful, sound, predictable, fast verifier for c and java.
>
> [2] Dafny: An automatic program verifier for functional correctness.
>
> [3] Vst-a: A foundationally sound annotation verifier.
>
> [4] Enchanting Program Specification Synthesis by Large Language Models using Static Analysis and Program Verification.
>
> [5] SLING: Using Dynamic Analysis to Infer Program Invariants in Separation Logic.
>
> [6] Learning Loop Invariants for Program Verification.
>
> [7] CLN2INV: LEARNING LOOP INVARIANTS WITH CONTINUOUS LOGIC NETWORKS.
>
> [8] Minif2f: a cross-system benchmark for formal olympiad-level mathematics.
>
> [9] Fimo: A challenge formal dataset for automated theorem proving.
>
> [10] Proofnet: Autoformalizing and formally proving undergraduate-level mathematics.
>
> [11] TRIGO: Benchmarking Formal Mathematical Proof Reduction for Generative Language Models.

---

> > ### Comment · Reviewer_rGBC · 2024-08-23
> >
> > Thanks for elaborating on the benchmark selection criteria. I did a quick look at the 12 programs taken from the real-world system LiteOS, which appear to be not quite realistic, as they basically iterate through one type of linked list namely, `LOS_DL_LIST`, (either forward or backward in ~5 lines) as follows:
> >
> > https://anonymous.4open.science/r/LIG-MM-NeurIPS24/dataset/System/lightos/changed_codes/los_list/iterator/iter.c
> >
> > ```C
> > struct LOS_DL_LIST {
> >     struct LOS_DL_LIST *pstPrev;
> >     struct LOS_DL_LIST *pstNext;
> > };
> > ```
> > I guess that these programs are dramatically simplified code, as in the real world, no one would perform the following operation (essentially no-op)
> >
> > ```C
> > struct LOS_DL_LIST *iter(struct LOS_DL_LIST *l)
> > {
> >     struct LOS_DL_LIST *p;
> >     p = l;
> >     while (p) {
> >         p = p->pstNext;
> >     }
> >     return l;
> > }
> > ```
> >
> > Also, it should be clarified that all four works highlighted above (AutoSpec, SLING, Code2Inv, CLN2INV) mainly make _technical_ contributions rather than dataset contributions.

---

> > > ### Author Response · Authors · 2024-08-24
> > >
> > > Thank you for your response. As you pointed out, real-world code often includes additional operations beyond the core functionality. In the case of the ``iter`` example, our modifications were based on a function that, apart from the iteration logic, primarily operates on a counter without affecting memory. Because these non-memory-related operations do not align with the focus of our study, we chose to omit them from our dataset and retained only the memory-related operations that are central to our analysis.
> > >
> > > Here are the original codes in LiteOS: (including the position of them)
> > > ```
> > > // https://gitee.com/LiteOS/LiteOS/blob/master/kernel/include/los_list.h line 422
> > > #define LOS_DL_LIST_FOR_EACH(item, list) \
> > >     for (item = (list)->pstNext;         \
> > >          (item) != (list);               \
> > >          item = (item)->pstNext)
> > >
> > >
> > > //https://gitee.com/LiteOS/LiteOS/blob/master/kernel/base/sched/sched_mq/los_sched.c line 109
> > > LOS_DL_LIST_FOR_EACH(queueNode, &priQueues[priority]) {
> > >         ++itemCnt;
> > > }
> > > ```
> > >
> > > Here are several similar examples in Linux: (including the position of them)
> > > ```
> > > //https://github.com/torvalds/linux/blob/master/include/linux/list.h line 686
> > > #define list_for_each(pos, head) \
> > > 	for (pos = (head)->next; !list_is_head(pos, (head)); pos = pos->next)
> > >
> > > //https://github.com/torvalds/linux/blob/master/drivers/net/ethernet/broadcom/genet/bcmgenet.c line 1561
> > > static int bcmgenet_get_num_flows(struct bcmgenet_priv *priv)
> > > {
> > > 	struct list_head *pos;
> > > 	int res = 0;
> > >
> > > 	list_for_each(pos, &priv->rxnfc_list)
> > > 		res++;
> > >
> > > 	return res;
> > > }
> > > ```
> > >
> > > We believe this selective inclusion helps maintain the relevance of our benchmark to memory manipulation tasks, ensuring that the evaluation is directly tied to the challenges our method aims to address.
> > >
> > > Regarding your second question: ``Also, it should be clarified that all four works highlighted above (AutoSpec, SLING, Code2Inv, CLN2INV) mainly make technical contributions rather than dataset contributions.`` We fully agree that these works have made significant technical contributions, which is precisely why we included them as baselines in our experiments. We have thoroughly discussed their loop invariant generation methods as related works on page 3 and page 13 (Appendix A.1) of our paper.
> > >
> > > Besides, our work also makes important technical contributions other than the proposed LIG-MM benchmark. We introduce a novel method, LLM-SE, which combines large language models with symbolic execution. This approach is a significant advancement in the field, offering a new way to tackle the challenges of loop invariant generation. Our method goes beyond existing approaches by leveraging the strengths of LLMs, symbolic execution, and multiple auxiliary techniques (self-supervised learning, data augmentation, interactive system). **As acknowledged by other reviewers (HX1p, 4YxC, EQmi)**, we consider our proposed framework LLM-SE to include pioneering contributions that address gaps in current techniques.
> > >
> > > We hope that our responses can address your concerns. We would be happy to provide further responses if you have any questions or need clarification.

---

### Author Rebuttal · Authors · 2024-08-15

We thank all reviewers for their kind reviews and helpful comments. We are delighted with their encouraging comments:

(1) The idea is novel and important to the whole community. (rGBC, HX1p, 4YxC, EQmi)

(2) The proposed benchmark is well-motivated and missing in existing works. (rGBC, HX1p, EQmi)

(3) The experiments conducted on the proposed multi-source benchmark demonstrate the performance of the proposed method compared to other popular baselines. (4YxC, EQmi)

(4) The paper is well-writen and easy to follow. (rGBC, HX1p, 4YxC)

During our rebuttal, we post detailed responses to each reviewer and try to answer their questions, ease their concerns, and further improve our paper. We thank the reviewers again for the valuable comments that are helpful in refining our paper.

Our work provides a new loop invariant generation benchmark specifically designed for program verification involving complex memory manipulations, along with a novel LLM+symbolic execution approach as a preliminary baseline. **Our benchmark includes the most various real-world programs with data structures**, including course homework programs from college students, programs from previous competitions/benchmarks, and programs from mutliple real-world software systems (Linux, GlibC, LiteOS, Zephyr).

In contrast, existing benchmarks primarily focus on numerical programs that do not involve any data structures, making them less applicable to the real world. We believe that our benchmark can fill this gap. This contribution is intended to advance the field by enabling more comprehensive testing and development of program verification, especially for real-world programs.

---

### Author Response · Authors · 2024-08-23

Dear reviewers,

Thank you for your valuable comments on our paper. We sincerely appreciate the time and effort you have invested in reviewing our work.

As the discussion phase is now halfway through, we hope that our responses have adequately addressed your concerns. If there are any additional questions or clarifications needed, we would be happy to provide further responses.

Best regards,

The authors

---

### Decision · Program_Chairs · 2024-09-26

**Decision:**

Accept (Poster)

**Comment:**

This paper proposes a benchmark for formal verification, wherein the objective is to generate loop invariants for programs that manipulate memory (e.g., linked lists). Although the data set is still quite small (over 300 programs), this set addresses a need for more realistic datasets, as many existing loop invariant benchmarks do not contain memory manipulations. Because many existing methods, particularly LLMs, fail to find such memory-dependent loop invariants, the authors propose a framework that interleaves symbolic execution with LLMs to great effect. The symbolic execution + LLM approach is shown to be effective, and the code is made available to the reviewers.  Both the dataset and the proposed method may stimulate more research on this topic.